# Point Set Multi-Level Aggregation Feature Extraction Based on Multi-Scale Max Pooling and LDA for Point Cloud Classification

**Guofeng Tong [1], Yong Li [1,*], Weilong Zhang [2], Dong Chen [3], Zhenxin Zhang [4], Jingchao Yang [2] and Jianjun Zhang [1]**

[1] College of Information Science and Engineering, Northeastern University, Shenyang 110819, China; tongguofeng@ise.neu.edu.cn (G.T.); 1800931@stu.neu.edu.cn (J.Z.)
[2] Hebei Jiaotong Vocational and Technical College, Shijiazhuang 050035, China; long2000489@sohu.com (W.Z.); yang_jing_chao@hotmail.com (J.Y.)
[3] College of Civil Engineering, Nanjing Forestry University, Nanjing 210037, China; chendong@njfu.edu.cn
[4] Beijing Advanced Innovation Center for Imaging Theory and Technology, Capital Normal University, Beijing 100048, China; zhangzhx@cnu.edu.cn
* Correspondence: liyong@stumail.neu.edu.cn

**Abstract:** Accurate and effective classification of lidar point clouds with discriminative features expression is a challenging task for scene understanding. In order to improve the accuracy and the robustness of point cloud classification based on single point features, we propose a novel point set multi-level aggregation features extraction and fusion method based on multi-scale max pooling and latent Dirichlet allocation (LDA). To this end, in the hierarchical point set feature extraction, point sets of different levels and sizes are first adaptively generated through multi-level clustering. Then, more effective sparse representation is implemented by locality-constrained linear coding (LLC) based on single point features, which contributes to the extraction of discriminative individual point set features. Next, the local point set features are extracted by combining the max pooling method and the multi-scale pyramid structure constructed by the point's coordinates within each point set. The global and the local features of the point sets are effectively expressed by the fusion of multi-scale max pooling features and global features constructed by the point set LLC-LDA model. The point clouds are classified by using the point set multi-level aggregation features. Our experiments on two scenes of airborne laser scanning (ALS) point clouds—a mobile laser scanning (MLS) scene point cloud and a terrestrial laser scanning (TLS) scene point cloud—demonstrate the effectiveness of the proposed point set multi-level aggregation features for point cloud classification, and the proposed method outperforms other related and compared algorithms.

**Keywords:** point cloud classification; multi-level point sets; multi-scale features; max pooling

## 1. Introduction

Recently, lidar sensors have been widely used in many fields. Classification of laser scanning point clouds is an important technology in the applications of automatic driving, intelligent city, mapping, and remote sensing [1–4]. Due to a variety of complex objects with different sizes and geometric structures in point clouds, accurate and efficient classification of point clouds becomes very challenging [5,6]. Therefore, the research on point cloud classification is of great significance for scene understanding and object perception.

A large number of point cloud classification approaches have been proposed over the past decade. Those classification methods can be mainly classified into two categories: single point-based

methods and point set-based methods. Generally, the single point-based methods mainly consist of neighborhood selection, feature extraction, and classifier for each single point classification [5–9]. Among them, the methods of neighborhood selection mainly use radius, cylindrical region, or K-nearest neighbor (KNN) [7,8] to construct the neighborhood. Feature extraction methods include low-level feature extraction, higher level feature extraction, and feature selection based on low-level features. The low-level features include normal vector and elevation feature [5,8], spin image [6,10], covariance eigenvalue feature [11], view feature histogram (VFH) [12], and clustered view feature histogram (CVFH) [13], among others. Higher level features are mainly extracted by manifold learning [9,14], low-rank representation [15], sparse representation [6,16], and so on [17,18]. The most popular classifiers mainly include linear classifiers [19], random forests [20], AdaBoost [21], and SVM (support vector machine) [22]. For example, Mei et al. [9] extracted color information, normal vector, spin image, and elevation features of each point using nearest neighbor points selected by radius. Then, the margin, the co-graph, and the label constraints were used for feature learning and selection. Finally, the linear classifier was used to classify all points. However, the features extracted by single point-based methods are usually not stable and lack the structure and the correlation information between local points, thereby decreasing accuracy and robustness of single point-based classification methods [6,16,23]. To solve the above problems, researchers proposed several point set-based classification methods. For these methods, points with the same attributes are grouped into a patch, from which the features can be derived to improve the robustness and the discrimination ability of feature expression. In this case, the basic processing unit is the segmented point sets, which shows a high resistance to noise and outliers and can help improve the accuracy of point cloud classification.

Currently, point set construction methods can be generally categorized into cluster-based methods [24–29], region growing-based methods [20,29], graph cut and raster image-based methods [6,16,30], model-based methods [31,32], content-sensitive and raster image-based methods [33], voxel-based methods [34], and neighborhood-based methods [35,36]. However, point set construction relies on the point cloud segmentation/clustering algorithms. It is also difficult to analyze the topological structure, and it is not always easy to select the effective segmentation/clustering method [36], especially for point cloud scenes contaminated with many scattered points. For example, the region growing segmentation algorithm is greatly influenced by the selection of seed points and an appropriate clustering criterion. As the growth criterion construction and the low-level features selection have huge impact on the point clouds segmentation, the region growing-based algorithms usually are less robust. Model-based segmentation methods can only be applied to specific model categories. Graph cut and raster image-based method [6,16] and content sensitivity and raster image-based method [33] need to project point clouds to two-dimensional raster images, which increases the computational difficulty and does not guarantee the discrepancies between point sets. Moreover, if the number of constructed layers is small, it causes under-segmentation, which does not contribute to extract stable and salient features at different levels. Although the cluster-based approach represents an adaptability to a certain extent, it usually depends on the Euclidean distance metric for clustering. In some complex scenes, different objects are too close each other, which makes the clustering algorithm inapplicable. Additionally, it is difficult to segment the point cloud objects of different scales based on a clustering algorithm. To obtain more representative point sets at different levels for different objects, we propose a multi-level point set construction method based on point cloud density and maximum point constraints within point set. The proposed method first uses the DBSCAN (density-based spatial clustering of applications with noise) [27] algorithm to coarsely segment the point cloud. After implementing DBSCAN, the K-means algorithm [24] is used to iteratively segment every large-scale point set to guarantee that the number of points in every point set is less than a threshold $T$. Thereby, small-scale point sets can be generated. Jointing the two clustering algorithms, we can effectively construct multi-level point sets of different sizes, i.e., large scale point sets and small scale point sets.

The point set features can be extracted followed by point set construction. Generally, point set features are extracted mainly by low-level features of point set [23], BoW (bag of word), and LDA

(latent Dirichlet allocation) [6], sparse coding and LDA [16], and convolutional neural networks [35]. For example, Xu et. al. [33] projected the point clouds onto the ground to form a raster image, and then content-sensitive constraints were used to segment the raster image into super-pixels. Next, the normalized segmentation method [30] based on exponential function was used to obtain different levels of point sets. The sparse representations of low-level features for each point were obtained. Afterwards, the multi-level point set features were constructed based on the LDA model. Finally, the point set was classified by AdaBoost classifier. This method achieved better classification performance than the compared methods using point-based features, which also directly proves the effectiveness of point set-based methods and the robustness of high-level features based on point sets.

In addition, for the construction of multi-level point set features, references [6,16,33] extracted higher level features using LDA or other methods based on the sparse representation of single-point features through dictionary learning. Considering that the local region of the point cloud features has a certain correlation, these methods do not take the local structure relationships into account during sparse representation. That is, only the point set global features constructed by the LDA model are utilized, and LDA-based point set features lack the local structure information in the point set. To solve this problem, we propose a point set multi-level aggregation feature extraction framework. We first introduce locality-constrained linear coding (LLC) [37] for sparse representation of single point features. Then, a multi-scale point set feature construction method based on max pooling is proposed to obtain the point set local features. Afterwards, the LDA-based features defined at different hierarchical point sets (called LLC-LDA) and hierarchical multi-scale max pooling point set features (called LLC-MP) are fused to construct point set multi-level aggregation features. The fusion features can achieve the effective description of global and local point set features, thereby enhancing the stability and the discrimination ability of point set features.

The specific flowchart of the proposed point cloud classification algorithm is shown in Figure 1. Firstly, a multi-level point sets construction method based on point cloud density and the maximum point number is used to generate multi-level point sets. Then, considering the point cloud expression of local geometry and shape information, the multi-scale covariance eigenvalue features and the spin image features are extracted for each single point. Next, combining multi-level point sets, the LLC-LDA and the LLC-MP features can be extracted based on dictionary learning and sparse representation of single point features. Afterwards, global and local features of the point set can be generated by fusing LLC-LDA and LLC-MP features. In addition, different levels and types of point set features are transferred to the point set space at the finest layer, and then the multi-level aggregation features of the point sets are constructed by different types of features fusion. Finally, the multi-level aggregation features are used to classify the point clouds by SVM classifier.

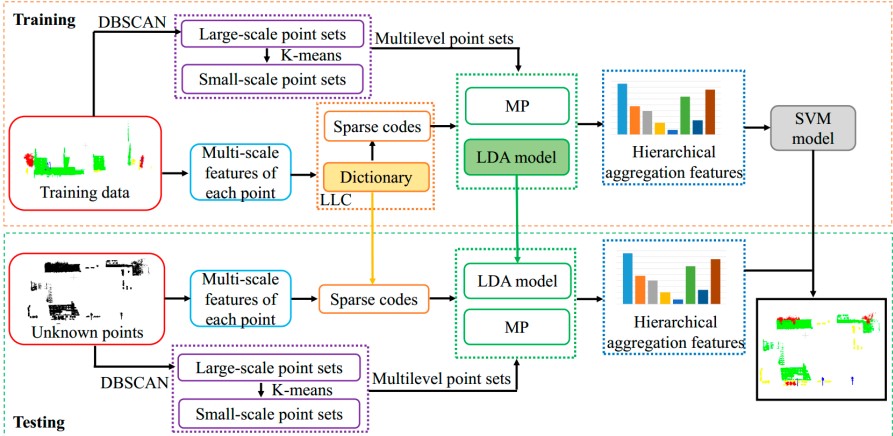

**Figure 1.** The flowchart of the proposed method. Note that LLC, LDA, MP, and SVM represent locality-constrained linear coding, latent Dirichlet allocation, multi-scale max pooling and support vector machine.

The main contributions of this paper are the following:

(1) A multi-level point sets construction method based on point cloud density and the maximum point number of point set is proposed, which can effectively construct different sizes and levels of point sets. The generation of the point sets does not require projection from the point cloud onto the two-dimensional grid, and it is possible to adaptively construct point sets of objects with different sizes. By controlling the maximum number of points in the point set, the fine-level point sets can be fully segmented. Different levels of point sets can contribute to construct effective point set features, which are more robust than single point features.

(2) A global feature extraction method called LLC-LDA of point set based on LLC and LDA models is proposed. LLC-based sparse coding considers the local relationships between individual point features and obtains more significant sparse representations than traditional sparse coding. Furthermore, the point set features constructed based on the LDA model are more stable and discriminative.

(3) A multi-level LLC-LDA and LLC-MP aggregation feature extraction and fusion method of the point set is proposed. The LLC-LDA mainly expresses the global features of the point set. The LLC-MP uses the spatial geometry to construct the point set features at multiple scales. That is, the features can reflect local features in point sets. Point set features at different levels are aggregated onto the point set at the finest level to generate the multi-level aggregation features of point sets. Once the local LLC-MP and the global LLC-LDA aggregation features are generated, we fused them together to obtain the final discriminative point set features.

## 2. Multi-Level Point Sets Construction

The point cloud classification based on the features of single point is susceptible to noise interference and the lack of relationship expression among points. To overcome the above problems, we extract point set features, followed by constructing multi-level point sets according to the constraints of density, position relationships, and point number. Different level point sets represent different scale information of the ground objects. Therefore, multi-level point sets can construct multi-level structures, which are more suitable for representation of objects with various sizes. For many constructed point sets [6], the number of point sets changes in a linear manner. The discrepancy of point sets between the adjacent levels might not be prominent enough. Thereby, the different level features of the same object and the same level features of different objects do not have a distinct difference. In fact, comprehensive descriptions of objects are generally achieved by multi-level features. The large-scale point sets can better represent the object at the global scale, while the small-scale point sets have the capability to describe the object at local and detailed scales. To obtain more representative point sets at different levels for describing different objects, we propose a multi-level point sets construction method based on the constraints of point cloud density and maximum point number of point set.

### 2.1. Large-Scale Point Set Construction Based on Point Cloud Density

Most existing segmentation methods are based on a fixed threshold of the size or the number of points, which is not suitable for various sizes of objects. Generally, outdoor scenes include many kinds of objects with various sizes and geometric shapes. Besides, there are some noises, outliers, occlusion, and data missing during the acquisition process. To get a reasonable number of segmentation units for all kinds of objects without knowing the number of object classes, DBSCAN is used for initial point cloud clustering. The specific steps of this algorithm are shown in references [23,24,27]. Here, different types of experimental scenes can be clustered according to the distribution of point clouds by DBSCAN algorithm.

As shown in the outdoor scene in Figure 2a, the point cloud can be roughly segmented at the first level clustering, as it is evident in Figure 2b. Compared with the ground truth shown in Figure 2d, it is observed that, in outdoor scenes, due to the similar point cloud density between cars and buildings, a small portion of cars and buildings are clustered together. That is, a phenomenon of

under-segmentation occurs. To make the structure and the class/label of point set more homogeneous, it is necessary to further segment the large-scale point set to achieve over-segmentation.

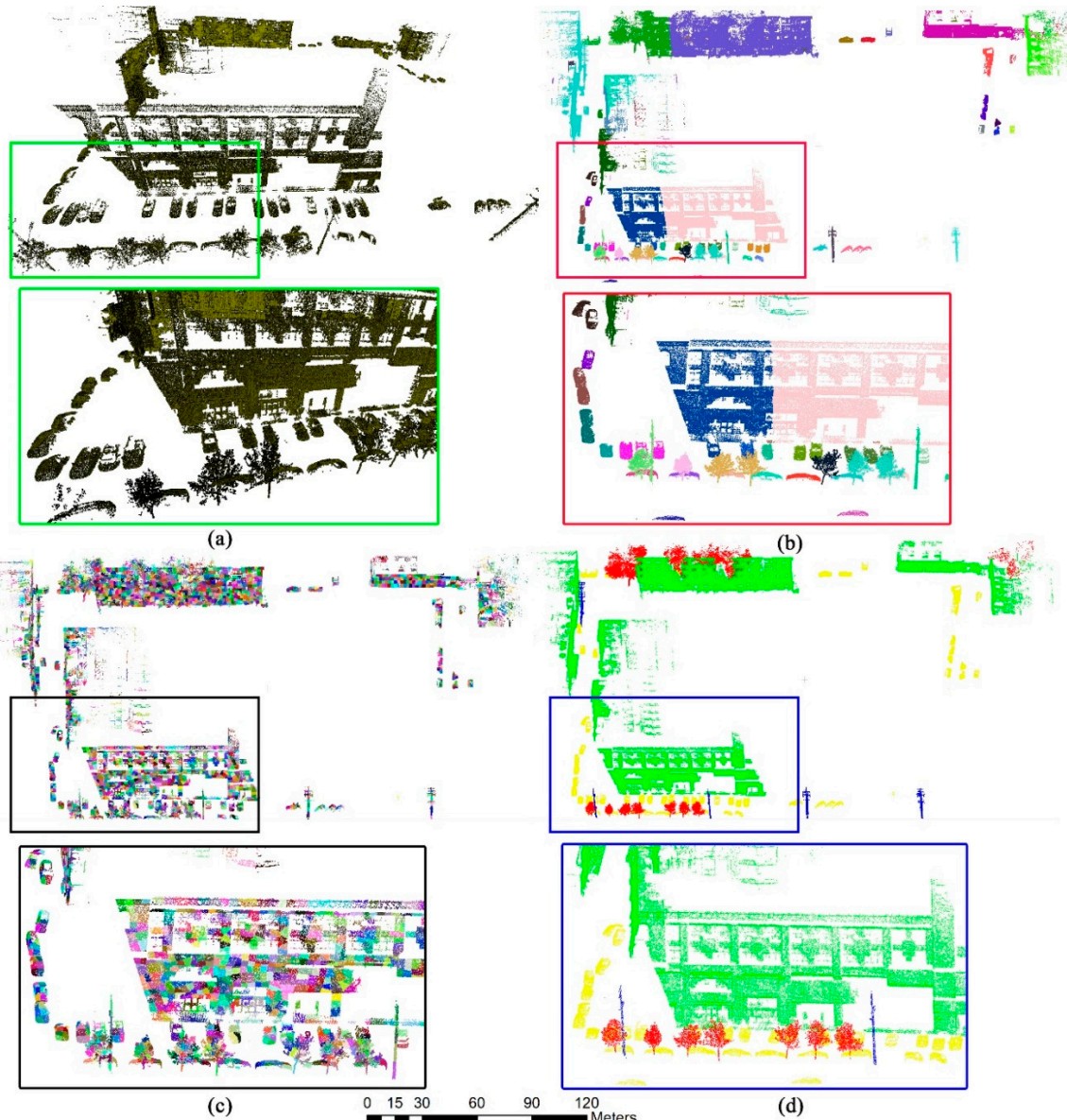

**Figure 2.** Multi-level point sets construction by combining density-based spatial clustering of applications with noise (DBSCAN) and *K*-means algorithms. (**a**) Original point clouds; (**b**) large-scale point sets; (**c**) small-scale point sets; (**d**) ground truth class label of each point. Please note that different colors represent different clusters in (**b**) and (**c**). A few colors are reused; as a result, different disjoint clusters may share the same color. In subfigure (**d**), the green points represent buildings, the red points represent trees, the yellow points represent cars, and the blue points represent utility poles.

*2.2. Adaptive Small-Scale Point Sets Construction Based on K-Means*

As shown in Figure 2b, the generated coarse point sets do not consider details and local distribution of the objects. In addition, the homogeneity of points within each point cloud cluster cannot be guaranteed. To overcome this deficiency, the K-means algorithm [24] is introduced to further segment coarse point sets. However, if K-means is directly used to segment the original point clouds, it needs more iterations and time cost. Therefore, we iteratively use K-means to over-segment coarse point sets clustered by DBSCAN with the threshold constraint. Here, we set K = 2, and K is the number

of cluster centers. This method can effectively cluster the coarse point sets into a large number of small-scale point sets with less than $T$ points. Afterwards, the majority labels of points within each point set have high probability of belonging to the same class. $T$ is a parameter that controls the size of the small-scale point sets. By this way, each coarse point set clustered by DBSCAN is further segmented to many over-segmented, smaller area/volume and homogeneous point sets. Almost all the points in the point set belong to the same class. The specific process of the segmentation method is shown as follows.

---

**Algorithm 1: K-Means-Based Adaptive Small-Scale Point Sets Construction Algorithm.**

---

**Input**: The coarse point sets obtained by DBSCAN $V = \{V_1, \ldots, V_N\}$ ($N$ is the number of point sets)
Parameters: The number of cluster centers $K$, The maximum point threshold in point set: $T$, maximum number of iterations $T_{iter}$.
**for** $I = 1{:}N$

**1:** An unlabeled point set $V_i$ is selected, and $K$ points are selected as the initial centroid $p_{i,c}^{(1)}$, c = 1,2, … ,K. (Make sure the distance between centroids is not too close).
**2: While** stop condition not met **do**
**2.1:** For the input point set, calculate the Euclidean distance of each point $p_{i,j}$ from the centroid $p_{i,c}^{(t)}$. Discriminate the category of each point according to the following equation, and obtain the point cloud clusters of each category $S_{i,c}^{(t)}$.

$$\begin{cases} p_{i,j} \in V_i \\ c^* = \underset{j}{arg\ min}\left(\| p_{i,j} - p_{i,c}^{(t)} \|\right), c, c^* = 1, 2, \ldots, K \end{cases}$$

**2.2:** Update the centroid of each category: $p_{i,c}^{(t+1)} = \frac{1}{n_c} \sum\limits_{j=1}^{n_c} p_{i,j} \Big| p_{i,j} \in S_{i,c}^{(t+1)}$ , $n_c$ is the points number of the $c$-th point cloud cluster.
**2.3:** The stop condition: $S_{i,c}^{(t+1)} = S_{i,c}^{(t)}, \forall c = 1, 2, \ldots, K$ or $t + 1 \geq T_{iter}$.
**end**
**3: for** $c = 1{:}K$ in $S_{i,c}^{(t+1)}$
**3.1: if** $n_c \leq T$
$S_v = S_{i,c}^{(t+1)}$, $v$ is the $v$-th output over-segmented point set.
$V = v{+}1$.
**3.2: else**
$S_\xi = S_{i,c}^{(t+1)}$, $s$ is the $s$-th point set with more than $T$ points.
$\xi = \xi{+}1$
**end**
**end**
**4:** Repeat **step 2** and **step 3** until the point cloud clusters $S_i^s = \varnothing$.
**end**
**Output:** Over-segmented point sets: $S_{clu} = \{S_1, \ldots, S_v\}$.

---

The small-scale point sets constructed by Algorithm 1 is shown in Figure 2c. Compared with Figure 2d, it can be seen that the points in the small-scale point sets almost belong to the same class, and the small-scale point sets reflect the characteristics of local region of the object. By making comparison with Figure 2b,c, we observe that the small-scale point sets actually describe the local region's geometries of the large-scale point sets in Figure 2b.

### 2.3. Multi-Level Point Sets Generation

As the point sets at a single scale cannot describe the object comprehensively, we construct point sets with multiple scales to effectively express the different size objects. A large-scale point set often describes the information of a large object or more objects belonging to the same category, while a small-scale point set can express the information of a small object or part of the object. We generate

the small-scale point sets at different levels by controlling the maximum point number threshold *T*. In addition, in order to obtain point sets with adjacent relationships and have more levels, a smaller threshold *T* is recommended to generate over-segmented point sets, which are then provided as input to the Mean-shift algorithm [24] to obtain more level point sets by tuning parameter of radius.

## 3. Multi-Level Point Set Features Extraction

This section mainly introduces the multi-level point set features extraction method. Firstly, we extract the features of each single point in the point cloud. Then, the multi-scale max pooling features and the LDA features of point set are constructed by sparse coding based on LLC.

### 3.1. Multi-Scale Single Point Features Extraction

With radius *R* as the support neighborhood, covariance eigenvalue features $\mathbf{F}_{cov}$ and spin image feature $\mathbf{F}_{si}$ of all points are extracted for each individual point *P*.

(1) Covariance eigenvalue feature

The covariance eigenvalue features [11] can be calculated by the Equations (1)–(3), and all the points in the support neighborhood are used to calculate the features. Each point in the point cloud can extract a set of six-dimensional covariance eigenvalue features within a neighborhood of radius *R*.

$$C_i = \frac{1}{k} \sum_{j=1}^{k} \left(P_j - p_i\right) \cdot \left(P_j - p_i\right)^T \tag{1}$$

$$\lambda_d = \lambda_d / \sum_{d=1}^{3} \lambda_d \tag{2}$$

$$F_{cov} = \left[ \sqrt[3]{\prod_{d=1}^{3} \lambda_d}, \frac{\lambda_1 - \lambda_3}{\lambda_1}, \frac{\lambda_2 - \lambda_3}{\lambda_1}, \frac{\lambda_3}{\lambda_1}, -\sum_{d=1}^{3} \lambda_d log(\lambda_d), \frac{\lambda_1 - \lambda_2}{\lambda_1} \right] \tag{3}$$

where *k* is the number of all points in the support neighborhood.

(2) Spin image feature

Spin image [6,16] can express the shape features of the adjacent region for a point in three-dimensional space. Due to the strong robustness to occlusion and background interference and the insensitivity to rigid transformation of spin image feature, it is widely used in point clouds registration and three-dimensional objects recognition [6,9,16,19,33,38–40]. Its specific extraction process is descried as follows.

For each point $p_i$, support neighborhood at radius *R* is $p_i^j$, $\left| p_i^j - p_i \right| \le R$, $j \in \left\{ 1, 2, \ldots, K_i^R \right\}$. The normal vector $\boldsymbol{n}_i^R$ of $p_i$ is first calculated. Then, $p_i$ and $\boldsymbol{n}_i^R$ are used as axes to construct a cylindrical coordinate system. The two-dimensional grid size of spin image is defined as $\mathcal{Z}_x \times \mathcal{Z}_y$, and the three-dimensional coordinates of the cylinder are projected onto the two-dimensional grid according to the following Equation:

$$\left(a_j^i, b_j^i\right) = \left( \sqrt{\| p_i^j - p_i \|^2 - \left(\boldsymbol{n}_i^R \cdot \left(p_i^j - p_i\right)\right)^2}, \boldsymbol{n}_i^R \cdot \left(p_i^j - p_i\right) \right), \left| p_i^j - p_i \right| \le R \tag{4}$$

where $a_j^i$ represents the X-axis coordinates of spin image constructed by three-dimensional point $p_i^j$ at point $p_i$, and $b_j^i$ represents the Y-axis coordinates of spin image constructed by three-dimensional

point $p_i^j$ at point $p_i$. According to spin image coordinate values, all the points in the neighborhood of $p_i$ falling into the corresponding grid are determined according to Equation (5).

$$grid_x = \left[ \frac{\mathcal{Z}_x a_j^i}{R} \right], grid_y = \left[ \frac{\mathcal{Z}_y}{2} \left( 1 - \frac{b_j^i}{R} \right) \right] \tag{5}$$

The number of points falling in each grid in the spin image is different. The intensity $I$ of each grid can be calculated according to the point number. Here, we build a $6 \times 6$ spin image for each point, which is a 36-dimensional feature vector (denoted by the symbol $\mathbf{F}_{si}$) for each point.

To make the local features expression more sufficient and robust, three support neighborhoods of different sizes are selected to construct multi-scale features. The size of supporting neighborhoods is determined radius $R$. In this article, parameter $R$ is selected as 0.2 m, 0.8 m, and 1.2 m. At each scale, each point can be represented by a 42-dimensional feature descriptor. Thereby, the multi-scale features of each point can be represented by a 126-dimensional feature descriptor, i.e., $\mathbf{F}_{\text{m-point}} = [\mathbf{F}_{cov\text{-}R1}, \mathbf{F}_{cov\text{-}R2},$ $\mathbf{F}_{cov\text{-}R3}, \mathbf{F}_{si\text{-}R1}, \mathbf{F}_{si\text{-}R2}, \mathbf{F}_{si\text{-}R3}]$.

The two quantities ($\mathbf{F}_{cov}$ and $\mathbf{F}_{si}$) do not have the same scale, which makes it biased towards one or the other. To solve it, we normalize all single point features $\mathbf{F}_{\text{m-point}}$ over each column. Each column represents each feature vector element of all points.

### 3.2. LLC-Based Dictionary Learning and Sparse Coding for Single Point Features

Since the original single point multi-scale features are low-level features, the expression of attributes for each single point is not significant. To make point cloud features more prominent and effective, BoW, low rank representation, manifold learning, and sparse coding are commonly used for feature selection [6,14,15]. Sparse coding, by learning a set of "super-complete" basis vectors to represent samples more efficiently, has significant advantages in dictionary learning and feature representation. For example, the better reconstruction performance and the sparse representation contribute to the salient feature extraction, and sparse features have better linear separability. However, according to reference [37], locality is more important than sparsity. Moreover, locality can guarantee the sparsity of coding, but the opposite is not true. The traditional sparse coding does not have a good locality. Generally, the neighboring points have the same or similar attributes, thereby, the local smoothness for sparse coding helps the features learning. To this end, the proposed method uses locality-constrained linear coding (LLC) to sparsely express point cloud features. Specific steps of LLC are described as follows.

The point cloud feature is normalized to $\mathbf{X} = [\ x_1, x_2, \ldots, x_N] \in \mathcal{R}^{D \times N}$, where $N$ is the number of points. $D$ is the dimension of each point feature. The dictionary of point cloud features is $\mathbf{B} = [\ b_1, b_2, \ldots, b_M] \in \mathcal{R}^{D \times M}$, and $M$ is the number of words in the dictionary. The sparse coding of $\mathbf{X}$ is $\mathbf{V} = [\ v_1, v_2, \ldots, v_N] \in \mathcal{R}^{M \times N}$. The traditional dictionary learning and sparse coding model is shown in Equation (6).

$$\min_{V, B} \sum_{i=1}^{N} \| X - BV \|^2 + \lambda |V| \tag{6}$$

$$s.t. \| b_j \| \leq 1, \forall j = 1, 2, \ldots, M$$

Considering the locality-constrained, the Equation (6) can be improved to construct the LLC model, which is Equation (7).

$$\min_{V, B} \sum_{i=1}^{N} \| x_i - Bv_i \|^2 + \lambda \| d_i \odot v_i \|^2 \tag{7}$$

$$s.t. \forall i, 1^T v_i = 1. \forall j, \| b_j \| \leq 1$$

where $\odot$ is the inner product of the element, and $\lambda$ is the constraint regular term parameter. $d_i \in \mathcal{R}^M$ is a local constraint condition, and it is defined as:

$$d_i = exp\left(\frac{dist(x_i, B)}{\sigma}\right) \tag{8}$$

We should note that, in Equation (8), $dist(x_i, B) = [dist(x_1, b_1), \dots, dist(x_i, b_M)]^T$, $dist(x_i, b_j)$ is the Euclidean distance of $x_i$ and $b_j$. $\sigma$ is a parameter that controls the range of the local region. In order to ensure that $V$ has sparsity and local smoothness, the element of $|v_i| < \varepsilon$ needs to be set to zero. To learn the optimal dictionary of point cloud features and the corresponding optimal sparse representation, we use the algorithm in reference [37] to optimize the objective function (6). During optimization, the initialized dictionary $B_{int}$ is first obtained by the K-means algorithm, wherein the number of words is $M$, i.e., $K = M$ in the K-means algorithm. For Equation (7), $V(B)$ is iteratively optimized by the coordinate descent method based on B($V$). Finally, the optimized dictionary $B$ and the corresponding sparse representation $V$ are obtained.

### 3.3. Multi-Level Point Set Features Construction

Single point features lack descriptions of the relationship between points, and they are sensitive to noise and outliers. We construct hierarchical point set features according to the different levels of point sets. The multi-level point set features mainly include two types: point set features based on LDA (LLC-LDA) and point set features based on multi-scale max pooling (LLC-MP).

#### 3.3.1. Point Set Features Extraction Based on LDA (LLC-LDA)

To obtain different types of high level features, we construct topic models by statistical features of each level point set. Based on the topic model, LLC-LDA features of each point set can be extracted. The specific construction steps are as follows:

First, the frequency of each word in each point set is counted based on the sparse representation matrix $V$ of LLC. The frequency of the $i$-th word in a point set is calculated according to Equation (9).

$$p(b_i|\theta, \beta) = \sum_{j=1}^{N_r} v_i^j \tag{9}$$

where $v_i^j$ represents the frequency of the $i$-th word for the sparse representation of $j$-th point in the point set. $N_r$ is the number of points in the point set, $\beta$ is a matrix with size $\ell \times M$, and $\ell$ is the number of latent topics. $\theta$ is a $\ell$-dimensional Dirichlet random variable, i.e., $\theta = [\theta_1, \dots, \theta_\ell]$, and $\theta_i$ is the probability of the $i$-th latent topic. Afterwards, the LDA model can be constructed as follows [41]:

$$p(B|\alpha, \beta) = \frac{\Gamma(\sum_i \alpha_i)}{\prod_i \Gamma(\sum_i \alpha_i)} \int \left(\prod_{i=1}^{j} \theta_i^{\alpha_i - 1}\right)\left(\prod_{m=1}^{M} \sum_{w_m} p(w_m|\theta)p(b_m|w_m, \beta)\right) d\theta \tag{10}$$

where $\alpha$ is the Dirichlet parameter, and the latent topic set is: $w = [w_1, \dots, w_M]$.

For Equations (10) and (11), the expectation maximum (EM) algorithm [16] is used to optimize $\alpha$ and $\beta$. Based on these two optimized parameters, the point set probability of each latent topic can be obtained. Subsequently, the point set feature is constructed based on the probability of all the latent topics. The LLC-LDA feature of the $l$-th point set on the $L$-th level can be expressed as follows:

$$F_{C_L^l}^{LDA} = [\theta_1, \dots, \theta_\ell] \tag{11}$$

### 3.3.2. Point Set Features Extraction Based on Multi-Scale Max Pooling (LLC-MP)

LLC-LDA features describe the global features of all points in point sets at each level. However, there is a certain structural relationship among points in point set. To fully express the attribute of the point set with local structure information, inspired by the structure of a space pyramid, we construct a multi-scale pyramid using spatial coordinates of each point set. Then, the max pooling method is used to extract nonlinear features of point sets at each scale. In the last step, the features of each scale are fused to obtain the position–feature space features of the point set. From another perspective, this method can construct smaller scale point sets and express the relationships of these smaller scale point sets (local regions) for the current level point set. The specific LLC-MP features extraction is as follows:

Given a point set, for the $s$-th ($s \in [1, P_s]$) scale space, $P_s$ is the number of scale spaces. The point set is divided into $K_s$ subspaces based on the spatial coordinates of the point set. Then, the different scale spaces of the point set can be constructed.

For the $s$-th scale, the point set max pooling features of $K_s$ subspaces can be calculated according to Equation (12).

$$f^{i,s} = \mathcal{F}\left(\overline{V_s}\right), \overline{V_s} = V_s{}', \overline{V_s} \in \mathcal{R}^{N_s \times M} \tag{12}$$

where $\mathcal{F}$ is the max pooling function. $V_s$ and $N_s$ are the sparse representation matrix of the $i$-th ($i \in [1, K_s]$) subspace point sets and the number of points in the point set, respectively. $f^{i,s}$ can be calculated according to Equation (13).

$$\begin{cases} \mathbf{f}^{i,s} = \left[f_1^i, .., f_j^i\right], j \in [1, M] \\ f_j^i = z^s \times \max\left\{\left|\overline{v}_{1j}\right|, \left|\overline{v}_{2j}\right|, \ldots, \left|\overline{v}_{N_s j}\right|\right\} \end{cases} \tag{13}$$

In different scales, due to the different number of points in each subspace, the features of different scales have different effects on the description of the point set. Besides, different information of the point set can be described in different subspaces. Therefore, the max pooling features of different scales should have different weights, i.e., $z^s$. In this paper, due to the small number of points in the finest layer point set, we only construct two scale subspaces.

The multi-scale max pooling features (LLC-MP) of point sets can be described as follows:

$$f_{MP} = f^{1,s} +, \ldots, + f^{i,s}, i \in [1, K_s], s \in [1, P_s] \tag{14}$$

The features can be normalized according to the Equation (15):

$$F_{MP} = \frac{f_{MP}}{\sum_{i=1}^{M} \left(f_{MP}\right)^2} \tag{15}$$

If $F_{MP}$ represents the $l$-th point set feature of the $L$-th level, LLC-MP can be expressed as $F_{C_L^l}^{MP}$.

## 4. Point Cloud Classification Based on Fusion of Multi-Level Point Set Features

Different types of features have different representations for the object attributes. Different levels of point set features have different descriptions for the object. In order to fully and effectively express the attributes of the object, we fuse different types and different levels of point set features. Taking the LLC-LDA point set feature as an example, the point set features at different levels can be aggregated by the coordinate of the points for different point sets. Generally, the point set feature space of the $L$-th layer (the finest layer) is used as the basic space for features aggregation. As shown in Figure 3, the point set features of the first layer and the second layer are transferred to the point set feature space of

the *L*-th layer for features aggregation. The LLC-LDA multi-level aggregation features of the *l*-th point set can be expressed as follows.

$$F_{C_L^l}^{ALDA} = \left[ F_{C_1^l}^{LDA}, F_{C_2^l}^{LDA}, \ldots, F_{C_L^l}^{LDA} \right] \tag{16}$$

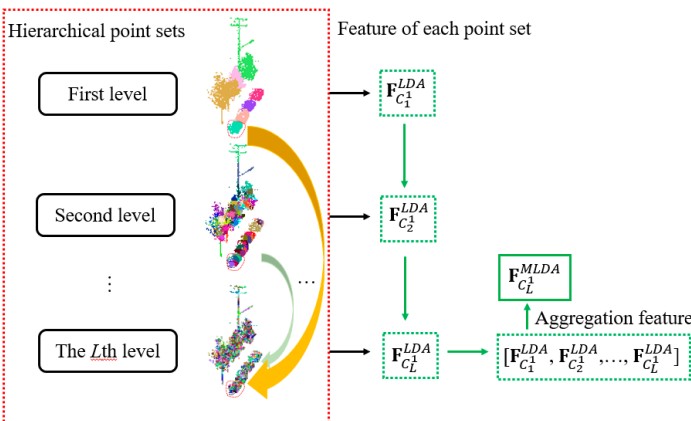

**Figure 3.** Point set multi-level aggregation features generation.

The above method can also be used to aggregate the LLC-MP multi-level point set features. Similarly, the LLC-MP multi-level aggregation feature of *l*-th point set can be expressed as follows:

$$F_{C_L^l}^{AMP} = \left[ F_{C_1^l}^{MP}, F_{C_2^l}^{MP}, \ldots, F_{C_L^l}^{MP} \right] \tag{17}$$

The LLC-LDA and the LLC-MP features of the point set reflect the global and the local features of the point set, respectively. To make full use of different types of features to classify the point sets, the two sets of features are fused. The fusion features of *l*-th point set are constructed as Equation (17). Afterwards, the point set can be classified according to the point set features.

$$F_{C_L^l} = \left[ F_{C_L^l}^{ALDA}, F_{C_L^l}^{AMP} \right] \tag{18}$$

In view of the excellent generalization ability and the relatively good adaptability to different data sizes of SVM, it is chosen as the classifier for the point cloud classification. In the experiment, we use the libsvm toolbox [42] to train and test the SVM model.

## 5. Experimental Results and Analysis

In this section, we carry out experiments on two different airborne laser scanning (ALS) point cloud scenes—a mobile laser scanning (MLS) point cloud scene and a terrestrial laser scanning (TLS) point cloud scene—to evaluate the effectiveness of the proposed algorithm. We conduct qualitative and quantitative analyses for the classification results to prove the advantages of the proposed method.

### 5.1. Experiment Data

To verify the effectiveness of the proposed algorithm, four different scenes are used for experiments. Among them, Scene1 and Scene2 are ALS point clouds provided by reference [16]. As shown in Figure 4, there are three categories of objects in the dataset, including large objects (buildings and trees) and small objects (cars). Scene3 is an MLS point cloud scene collected by a backpacked mobile mapping robot [43]. As shown in Figure 5, Scene3 contains four categories of objects, i.e., cars, poles, buildings,

and trees. Scene4 is a TLS point cloud scene provided by reference [9]. As shown in Figure 6, Scene4 contains pedestrians, cars, buildings, and trees. The point clouds of Scene1, Scene2, and Scene4 can be download at the website (http://geogother.bnu.edu.cn/teacherweb/zhangliqiang/). The ground (natural and artificial ground) points of these four scenes are manually filtered out using the open source tool Cloudcompare (http://www.couldcompare.org/). The details of different point cloud collection systems are shown in Table 1. The specific number of points in four scenes is shown in Table 2. The training set and the testing set of each scene are shown in Figures 4–6.

**Table 1.** The characteristics of collection systems and point clouds.

| Type | ALS | MLS | TLS |
|---|---|---|---|
| **Scenes** | **Scene1/Scene2** | **Scene3** | **Scene4** |
| Scanners | Leica ALS50 system | Backpacked mobile mapping robot (Omni SLAM$^{TM}$) [43] | RIEGL MS-Z620 |
| Range | A mean flying height of 500 m above ground and a 45° field of view | 0–100 m / field of view: 360° × 360° | 2–2000 m/ Horizontal and vertical angle spacing 0.57° |
| Accuracy/Precision | 150 mm/80 mm | 50 mm/30 mm | 10 mm/5 mm |
| Characteristic | The average strip overlap was 30%. Buildings with different roof shapes, e.g., flat and gable roofs, are surrounded by trees and cars. There are buildings with different heights, dense complex trees, and cars on the roads. The classes are unbalanced. | Buildings have varied densities, shapes, and sizes. Other pole-like objects (trees and poles) and cars are connected and mixed together. There are certain degree of noise and outliers scattered in this point clouds. Less affected by the distance changing. The classes are unbalanced. | The density of the point cloud varies according to the distance from the objects to the scanner. Trees are different shapes and densities. Many objects in this scene are incomplete, and many noise points distributes in this scene. The classes are unbalanced. |
| Point density | approximately 20–30 points/m$^2$ | approximately 100–180 points/m$^2$ | approximately 50–250 points/m$^2$ |
| Area | ~(237.7 m × 58.1 m)/ ~ (334.6 m ×0.5 m) | ~ (151.7 m × 178.3 m) | ~ (107.1 m × 79.9 m) |
| Scene type | Residential/Urban, Tianjin | Downtown, Shenyang | Campus, Beijing |

**Table 2.** The statistics of the training and the testing datasets for four scenes. Note that each number in the table represents the point number.

| | Training set Points | | | | | Test set Points | | | | |
|---|---|---|---|---|---|---|---|---|---|---|
| | Tree | Building | Car | Pole | Pedestrian | Tree | Building | Car | Pole | Pedestrian |
| Scene1 | 68,802 | 37,128 | 5380 | 0 | 0 | 213,990 | 200,549 | 7816 | 0 | 0 |
| Scene2 | 39,743 | 64,952 | 4,584 | 0 | 0 | 73,207 | 156,186 | 7409 | 0 | 0 |
| Scene3 | 35,078 | 140,164 | 15,936 | 5641 | 0 | 49,359 | 172,311 | 56,889 | 3711 | 0 |
| Scene4 | 125,610 | 45,341 | 1722 | 0 | 3087 | 178,391 | 13,906 | 48,759 | 0 | 16,381 |

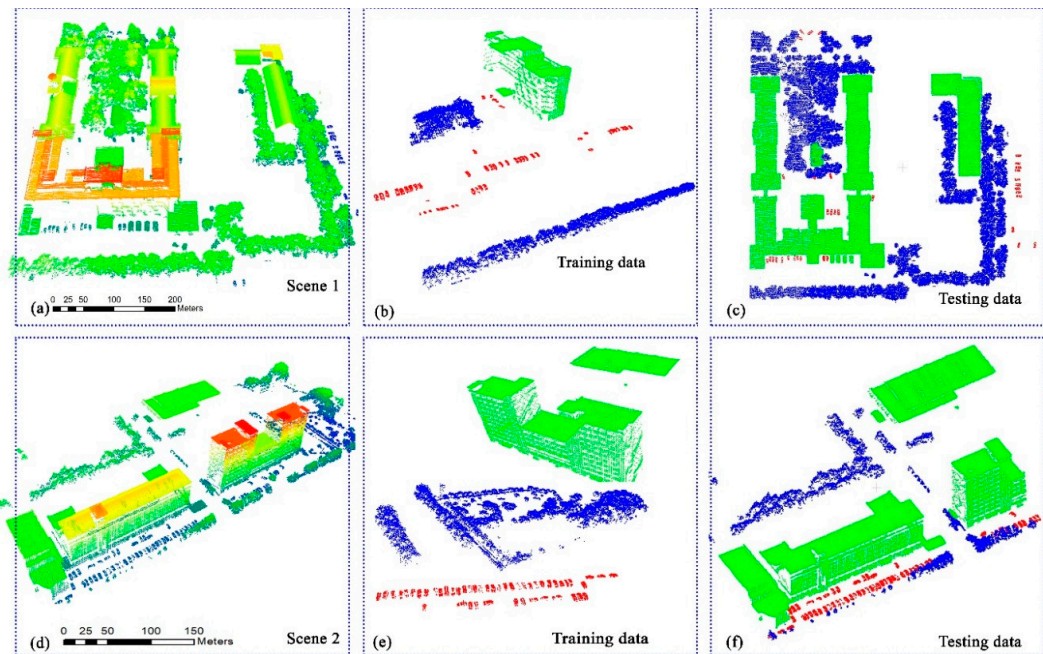

**Figure 4.** Airborne laser scanning (ALS) point cloud data. (**a**,**d**) are Scene1 and Scene2, respectively. (**b**,**e**) are the selected points with semantic labels from (**a**,**d**) for training. The testing data are shown in subfigures (**c**,**f**). Please note that the point clouds in (**a**,**d**) are rendered according to point clouds' elevation, and other colors represent the semantic information, i.e., blue = trees, green = buildings, and red = vehicles.

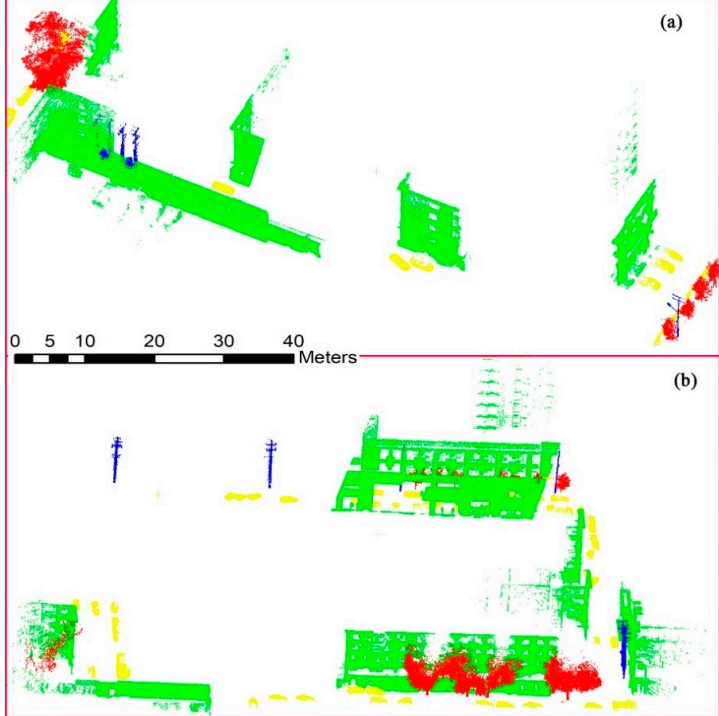

**Figure 5.** Scene3 mobile laser scanning (MLS) point clouds. (**a**) Training data, (**b**) testing data. Note that green, red, yellow, and blue points represent buildings, trees, cars, and poles, respectively.

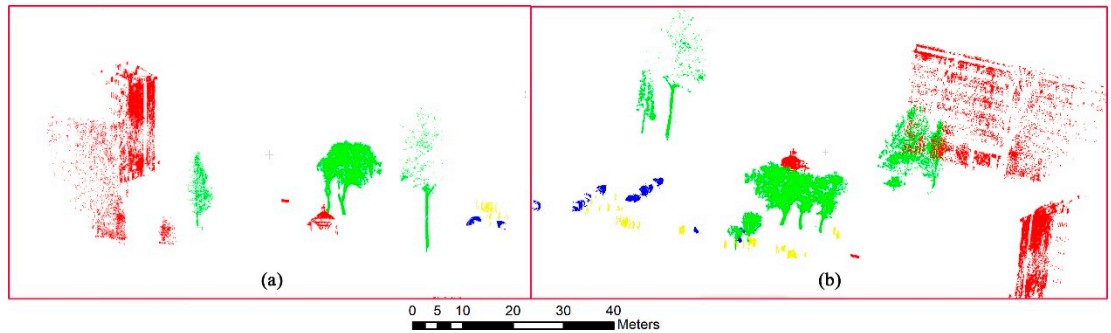

**Figure 6.** Scene4 terrestrial laser scanning (TLS) point clouds. (**a**) Training data, (**b**) testing data. Note that green, red, yellow, and blue points represent trees, buildings, pedestrians, and cars, respectively.

In our experiments, the proposed algorithm is implemented based on Microsoft Visual C++ (embedding PCL1.8.0) and MATLAB 2017b. All the experiments are run on a personal computer equipped with a 4.20 GHz Intel Core i7–7700k CPU, 24 GB of main memory. The average training time of four scenes is about 16.5 min, and the average testing time of four scenes is 2.3 min. In order to evaluate the performance of the proposed algorithm more comprehensively and effectively, we use Precision/Recall and $F_1$-score to evaluate the classification performance of each category. Overall accuracy (OA), mean Intersection over Union (mIoU), Kappa, and $mF_1$ are used to evaluate the overall classification performance of each scene. Here, Precision is the ratio of correctly predicted positive points to the total predicted positive points. Recall is the ratio of correctly predicted positive points to all points in the positive class. OA is the ratio of correctly predicted points to the total points. $F_1$-score is defined as: $F_1$-score = 2 × (Recall × Precision)/(Recall + Precision). $mF_1$ is computed by averaging all classes of the $F_1$-scores [23]. More details of these metrics are presented in reference [34,35].

$$mIoU = \sum_{i=1}^{N_c} \frac{h_{ii}}{h_{ii} + \sum_{i \neq j} h_{ij} + \sum_{j \neq i} h_{ji}} \tag{19}$$

$$Kappa = \frac{OA - \rho}{1 - \rho}, \rho = \frac{\sum_{j=1}^{N_c} \sum_{i=1}^{N_c} (h_{ij} \times h_{ji})}{N \times N} \tag{20}$$

where $H = \left[ h_{ij} \right]_{N_c \times N_c}$ is a confusion matrix, $h_{ij}$ is a number of points from ground-truth class $i$ predicted as class $j$. $N_c$ is the number of categories. $N$ is the number of all the points in the point cloud.

*5.2. Comparisons*

To highlight the performance of the proposed algorithm, we select 13 methods for comparisons. The characteristics of these comparison methods are shown in Table 3. **Method 1(LLC-LDA-SVM):** LLC-LDA-SVM is proposed in this paper, which extracts the multi-level point sets and single point features by our method. LLC is used to learn the dictionary. Then, the LLC-LDA point set features are aggregated to construct multi-level point set features. Finally, the LLC-LDA aggregation features are used for point cloud classification based on SVM. **Method 2 (LLC-MP-SVM):** LLC-MP-SVM is proposed in this paper, which is similar to Method 1. For Method 2, the LLC-LDA point set features are replaced by LLC-MP point set features. **Method 3 (DKSVD):** (Discriminative K-SVD): It uses DKSVD [44] to classify the point clouds based on the fusion features of multi-scale $F_{SI}$ and $F_{cov}$. In our experiment, the dictionary word is set to 128, and the regular term parameter is set to 0.1. **Method 4 (LCKSVD1)** and **Method 5 (LCKSVD2)** (Label Consistent K-SVD): They use LCKSVD1 [45] and LCKSVD2 [45] to classify the point clouds based on the fusion features of multi-scale $F_{SI}$ and $F_{cov}$, respectively. For LCKSVD1 and LCKSVD2, the number of dictionary words is selected from the set {64, 128, 256, 512}. The regular terms parameters are chosen from the set {0.001, 0.01, 0.1, 1, 10}. In our

case, when we choose the values of 512 and 0.01, LCKSVD1 and LCKSVD2 can both get the optimal results for all scenes. **Method 6 (MSF-SVM)** (multi-scale fusion features classified by SVM) [42]: It is a point-based method, which employs SVM to classify the point clouds based on the fusion features of multi-scale $F_{SI}$ and $F_{cov}$. **Mothod 7 (ECF-SVM)** (elevation and covariance eigenvalues features classified by SVM): A compared method proposed in reference [5]. It uses multi-scale elevation $F_z$ and covariance eigenvalues features $F_{cov}$ of single point to classify the point clouds. Afterwards, the classification results are optimized by multi-scale neighbors. **Method 8 (JointBoost)** [37]: Each point feature is constructed by geometry, strength, and statistics information. The JointBoost is used for features selection and point clouds classification followed by each point feature constructed by geometry, strength, and statistics features. **Method 9 (AdaBoost)**: It is a compared method proposed in reference [16]. This method uses AdaBoost to classify the point clouds based on the fusion features of multi-scale $F_{SI}$ and $F_{cov}$. **Method 10 (BoW-LDA)** [6]: It uses graph cut and linear transform to construct multi-level point sets. Then, K-means is employed for dictionary learning based on $F_{SI}$ and $F_{cov}$ fusion features. Afterwards, the multi-level point sets features can be constructed based on LDA model for point cloud classification. **Method 11 (DD-SCLDA)** (discriminative dictionary based sparse coding and LDA) [39]: Based on graph cut and exponential transformation, multi-level point sets can be constructed by DD-SCLDA. Fusion features of multi-scale $F_{SI}$ and $F_{cov}$ are used to learn the dictionary in DD-SCLDA. Then, a DD-SCLDA model is constructed to extract multi-level point set features. Finally, the point set features are aggregated for point clouds classification based on AdaBoost. **Method 12 (SC-LDA-MP)** [16,46]: Based on the multi-level point sets and single point features extracted by our method, the traditional sparse coding (SC) method is used to learn the dictionary. Then, the SC-LDA (sparse coding and LDA) point set features and the SC-MP (sparse coding and max pooling) point set features are fused as SC-LDA-MP (sparse coding, LDA, and max pooling) to classify the point clouds. Here, the number of dictionary words, the number of latent topics, and the point sets of the SC-LDA-MP are the same as our method. The dictionary learning initialization method is the same as well. The other parameters of the traditional sparse coding method are set as the optimal parameters given in reference [16]. Method 13 (PointNet) [17]: It is a deep learning network based on a multilayer perceptron, which is regarded as a baseline in reference [18]. The network can extract the features of each point and classify the point clouds. Here, we give the classification results of Scene1 and Scene2 based on PointNet. For the above 13 methods, $F_{SI}$ and $F_{cov}$ are features described in Section 3.1.

**Table 3.** Main characteristics of the proposed algorithm and other comparison algorithms.

| Method | Point Set Construction | Point Cloud Features | Dictionary and Features Expression | Classifier |
|---|---|---|---|---|
| Our method | Multi-level clustering | $F_{SI} + F_{cov}$ | LLC, Point set features fusion of LLC-LDA and LLC-MP | SVM |
| LLC-LDA-SVM | Multi-level clustering | $F_{SI} + F_{cov}$ | LLC, Point set features of LLC-LDA | SVM |
| LLC-MP-SVM | Multi-level clustering | $F_{SI} + F_{cov}$ | LLC, Point set features of LLC-MP | SVM |
| DKSVD [44] | Single point | $F_{SI} + F_{cov}$ | DKSVD, Dictionary-based sparse representation | Linear classifier |
| LCKSVD1 [45] | Single point | $F_{SI} + F_{cov}$ | LCKSVD1, Sparse representation based on saliency dictionary | Linear classifier |
| LCKSVD2 [45] | Single point | $F_{SI} + F_{cov}$ | LCKSVD2, Sparse representation based on saliency dictionary | Linear classifier |
| MSF-SVM [42] | Single point | $F_{SI} + F_{cov}$ | No dictionary, single point features | SVM |
| ECF-SVM [5] | Single point | $F_z + F_{cov}$ | No dictionary, single point features | SVM |

| Method | Point Set Construction | Point Cloud Features | Dictionary and Features Expression | Classifier |
|---|---|---|---|---|
| JointBoost [38] | Single point | Geometry, strength, and statistical features | No dictionary, single point features | JointBoost |
| AdaBoost [16] | Single point | $F_{SI} + F_{cov}$ | No dictionary, single point features | AdaBoost |
| BoW-LDA [6] | Graph cut and linear transformation | $F_{SI} + F_{cov}$ | K-means, Point set features of LDA | AdaBoost |
| DD-SCLDA [39] | Graph cut and exponential transformation | $F_{SI} + F_{cov}$ | LCKSVD, Point set features of DD-SCLDA | AdaBoost |
| SC-LDA-MP [16,46] | Multi-level clustering | $F_{SI} + F_{cov}$ | SC, Point set features fusion of SC-LDA and SC-MP (SC-LDA-MP) | SVM |
| PointNet [17] | Point cloud block | Point features based on deep learning | No dictionary, multi-layer perceptron (MLP) | Softmax |

Notes: DD-SCLDA: discriminative dictionary based sparse coding and LDA; BoW: bag of word; MSF-SVM: multi-scale fusion features classified by SVM; ECF-SVM: elevation and covariance eigenvalues features classified by SVM; SVM: support vector machine; DKSVD: discriminative K-SVD; LCKSVD: label consistent K-SVD.

### 5.2.1. ALS Point Clouds

In this part, Scene1 and Scene2 are tested. The details of the training set and the test set for each scene are shown in Table 2. Table 4 gives the classification results of different methods shown in Table 3. Because the source codes of some compared methods are not provided, we cannot get the results of these methods. For unbiased comparisons, some metric values and results of some methods are not compared in Table 4 and Figures 7 and 8.

From the results listed in Table 4, we have the following observations:

(1) Our method achieves 96.7%/95.3%, 77.9%/76.0%, 93.6%/90.1%, and 85.4%/84.3% with regard to OA, mIoU, Kappa, and $mF_1$ on Scene1 and Scene2, which maintains the highest evaluation metric values and demonstrates the advantages of the proposed method.

(2) For LLC-LDA-SVM and LLC-MP-SVM, these two methods cannot achieve good performance on cars, and the extracted features are not robust for classification, especially for small objects. However, our method fuses two features (the global features of the point set and the local distribution features of the point set) to extract more discriminative features for the point sets representation and classification. It demonstrates that the introduced LLC-MP features and the fusion with the LLC-LDA features are effective for point cloud classification.

(3) Methods 3–8 are point cloud classification methods based on single point features, while other methods classify point clouds based on point set features. From the $F_1$-score of each category classification and the $mF_1$ of all categories classification in Table 4, it can be seen that classification methods based on point set features can obtain higher $F_1$-score and $mF_1$ in most cases compared to classification methods based on single point features, i.e., point set features are more robust than single point features for point cloud classification. The five point-based methods, i.e., DKSVD, LCKSVD1, LCKSVD2, MSF-SVM, and AdaBoost, are not robust for most categories classification. The discriminant features extracted/learned by these methods are not ideal, especially for small sample objects. Although ECF-SVM and JointBoost can achieve better performance in point-based methods, the anti-noise ability of these methods still needs to be improved. In addition, BOW-LDA and DD-SCLDA construct more than two levels of point sets. The point sets constructed by these two methods are not rich enough to express different scale objects and different regions of the objects. In our experiment, our method only constructed two levels of point sets, but the proposed method outperforms the comparison methods.

(4) As shown in Table 3, Methods 1–6, Methods 9–12, and the proposed method use similar single point features. It can be seen from the classification results in Table 4 that learning and representation of different point cloud features and classifiers have a great influence on the performance of point

cloud classification. As shown in Table 4, the combination of feature learning, feature expression, and classifier in our method has better performance than other compared methods in most metrics.

(5) Compared with DKSVD, LCKSVD1, and LCKSVD2, our method can achieve at least 15.9%, 30.8%, 32.7%, and 28.8% higher than these three compared methods with regard to OA, mIoU, Kappa, and $mF_1$ on Scene1/Scene2. It demonstrates that the classification performance of our method has obvious advantages in the overall classification metrics. This proves that the point set features constructed by single point features dictionary learning and sparse representation are more discriminative than the single point features constructed by dictionary learning and sparse representation.

(6) The OA and the $mF_1$ of our method are at least 31.4% and 38.0% higher than the deep learning method of PointNet. It demonstrates that our method obviously outperforms PointNet. It also proves that, when the number of training samples is relatively small, the deep learning method, i.e., PointNet, cannot extract effective point cloud features for classification. It should be noted that the machine learning method is relatively more efficient than the deep learning method when the number of training samples is small.

(7) By making a comparison between SC-LDA-MP and the proposed method, it can be seen that OA, mIoU, Kappa, and $mF_1$ of our method on Scene1/Scene2 are 1.1%/0.3%, 6.7%/2.8%, 2.3%/0.8%, and 6.5%/2.6% higher than SC-LDA-MP. It proves that the introduced LLC plays a positive role in the dictionary learning and the sparse representation for the point cloud classification. It also demonstrates the introduced LLC is effective for the discriminative improvement of multi-level aggregation features of point sets.

**Table 4.** Classification results of Precision/Recall, overall accuracy (OA), mean Intersection over Union (mIoU), Kappa and $F_1$-score (%) on Scene1 and Scene2. The best results are highlighted in bold. The symbol "-" stands when the corresponding values are not given.

| Scene1 | Tree | Building | Car | OA | mIoU | Kappa | $F_1$-score | $mF_1$ |
|---|---|---|---|---|---|---|---|---|
| Our method | 96.6/**97.7** | **98.6**/96.0 | 47.9/**87.0** | **96.7** | **77.9** | **93.6** | **97.2**/**97.3**/61.8 | **85.4** |
| LLC-LDA-SVM | 97.6/86.7 | 89.0/**98.6** | 24.6/18.0 | 92.8 | 65.8 | 86.3 | 91.8/93.6/20.8 | 73.1 |
| LLC-MP-SVM | 98.3/85.7 | 88.2/**98.6** | 37.9/39.5 | 87.3 | 59.2 | 75.6 | 91.6/93.1/38.7 | 69.5 |
| DKSVD | 85.4/71.3 | 76.4/88.1 | 1.6/1.6 | 79.2 | 44.6 | 59.0 | 77.7/81.8/1.6 | 53.7 |
| LCKSVD1 | 84.3/59.2 | 71.1/86.7 | 2.6/10.1 | 72.8 | 39.8 | 47.6 | 69.6/78.1/4.1 | 50.6 |
| LCKSVD2 | 88.2/70.7 | 77.0/90.3 | 3.0/4.4 | 80.2 | 45.9 | 60.9 | 78.5/83.1/3.6 | 55.1 |
| MSF-SVM | 91.0/82.3 | 84.0/93.1 | 0.0/0.0 | 87.0 | 51.8 | 74.2 | 86.4/88.3/0.0 | 58.2 |
| ECF-SVM | **99.2**/84.9 | 86.8/99.3 | **99.9**/42.7 | 91.9 | – | – | 91.5/92.7/59.8 | 81.3 |
| JointBoost | 89.7/98.1 | 97.9/89.1 | 65.2/46.6 | 92.9 | – | – | 93.7/93.3/54.4 | 80.5 |
| AdaBoost | 85.7/92.9 | 92.0/83.8 | 56.9/54.7 | 87.9 | – | – | 89.2/87.7/55.8 | 77.6 |
| BOW-LDA | 94.8/93.8 | 93.5/92.3 | 41.2/66.7 | 92.6 | – | – | 94.3/92.9/50.9 | 79.4 |
| DD-SCLDA | 93.1/96.0 | 95.2/92.6 | 73.3/62.2 | 93.7 | – | – | 94.5/93.9/**67.3** | 85.2 |
| SC-LDA-MP | 98.3/93.7 | 93.8/98.5 | 55.6/37.3 | 95.6 | 71.2 | 91.3 | 95.4/95.7/43.4 | 78.9 |
| PointNet | 65.1/93.7 | 95.6/19.5 | 93.4/8.2 | 65.3 | – | – | 76.8/32.4/15.1 | 41.4 |
| **Scene2** | **Tree** | **Building** | **Car** | **OA** | **mIoU** | **Kappa** | **$F_1$-score** | **$mF_1$** |
| Our method | 93.4/92.7 | 99.2/97.5 | 52.4/**73.9** | **95.3** | **76.0** | **90.1** | 93.1/**98.4**/61.3 | **84.3** |
| LLC-LDA-SVM | **93.9**/90.6 | 97.7/97.3 | 48.9/68.8 | 94.3 | 73.6 | 88.1 | 92.2/97.5/57.2 | 82.3 |
| LLC-MP-SVM | 76.2/93.2 | 99.1/88.3 | 49.4/53.0 | 88.7 | 64.7 | 77.2 | 83.8/93.4/51.2 | 76.1 |
| DKSVD | 66.0/79.5 | 88.2/83.2 | 4.4/0.8 | 79.4 | 44.0 | 56.7 | 72.1/85.6/1.4 | 53.0 |
| LCKSVD1 | 47.1/79.6 | 87.1/54.3 | 5.0/10.2 | 60.7 | 31.9 | 30.6 | 59.2/66.9/6.7 | 44.3 |
| LCKSVD2 | 67.7/76.3 | 88.2/83.5 | 9.7/8.4 | 78.8 | 45.2 | 56.0 | 71.7/85.8/9.0 | 55.5 |
| MSF-SVM | 77.1/81.5 | 88.7/90.6 | 0.0/0.0 | 84.9 | 49.0 | 66.9 | 79.3/89.6/0.0 | 56.3 |
| ECF-SVM | 83.2/92.9 | 98.5/92.8 | 62.6/65.7 | 92.0 | – | – | 87.8/95.6/64.1 | 82.5 |
| JointBoost | 86.8/91.2 | 96.8/95.5 | 44.1/34.8 | 92.2 | – | – | 88.9/96.1/38.9 | 74.6 |
| AdaBoost | 73.9/91.2 | 93.6/88.2 | 29.5/25.4 | 87.2 | – | – | 81.6/90.8/27.3 | 66.6 |
| BOW-LDA | 90.3/93.9 | 97.6/96.5 | 49.4/42.0 | 94.1 | – | – | 92.1/97.0/45.4 | 78.2 |
| DD-SCLDA | – | – | – | – | – | – | – | – |
| SC-LDA-MP | 90.8/**94.4** | 98.0/**97.6** | **66.4**/46.4 | 95.0 | 73.2 | 89.3 | 92.6/97.8/54.6 | 81.7 |
| PointNet | 78.2/91.4 | 90.4/20.1 | 87.1/12.3 | 41.3 | – | – | 84.3/32.9/21.6 | 46.3 |

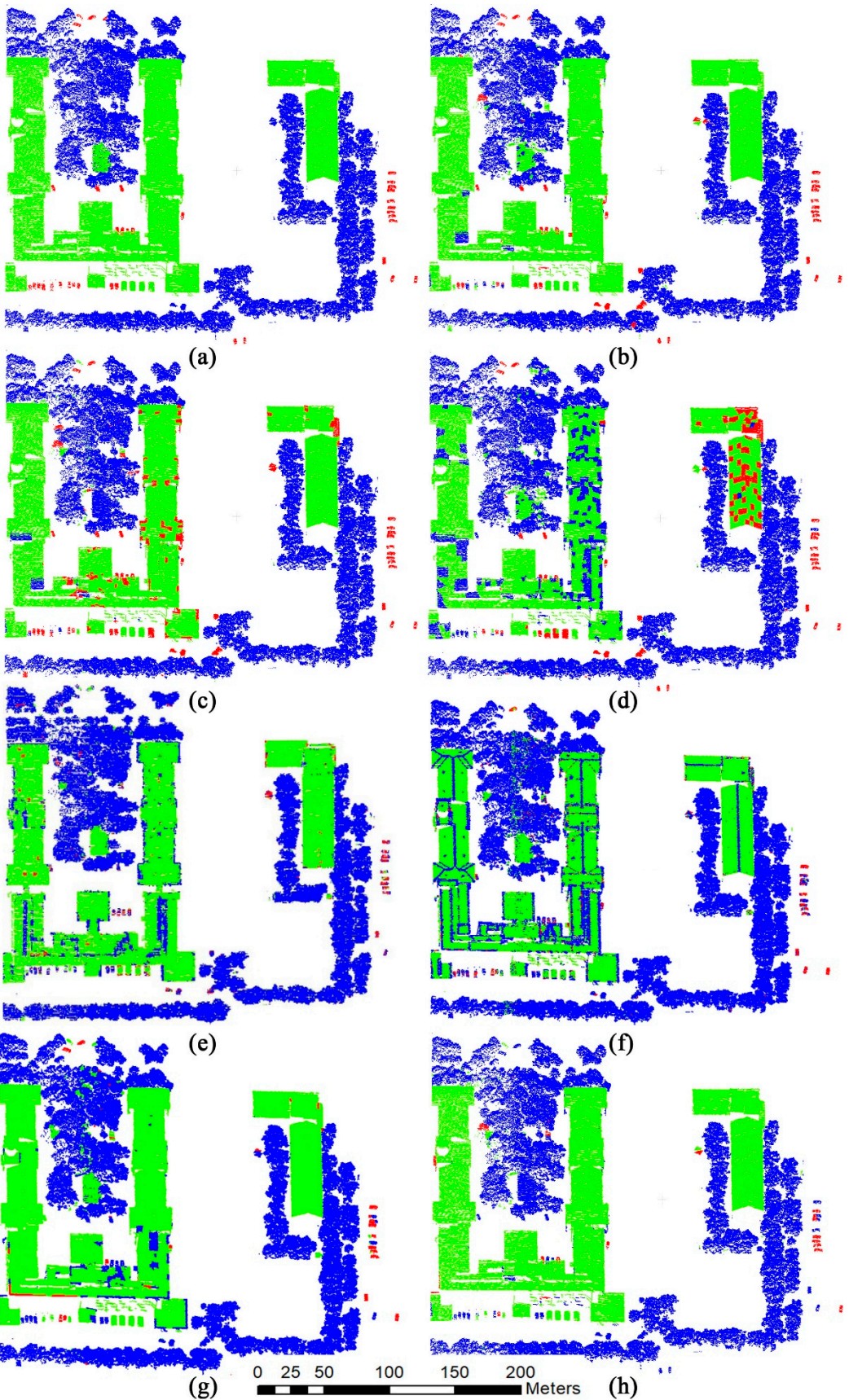

**Figure 7.** The classification results of Scene1. (**a**) Ground truth; (**b**) our method; (**c**) LLC-LDA-SVM; (**d**) LLC-MP-SVM; (**e**) JointBoost; (**f**) AdaBoost; (**g**) BOW-LDA; (**h**) SC-LDA-MP. Note that red, green, and blue points represent cars, buildings, and trees, respectively.

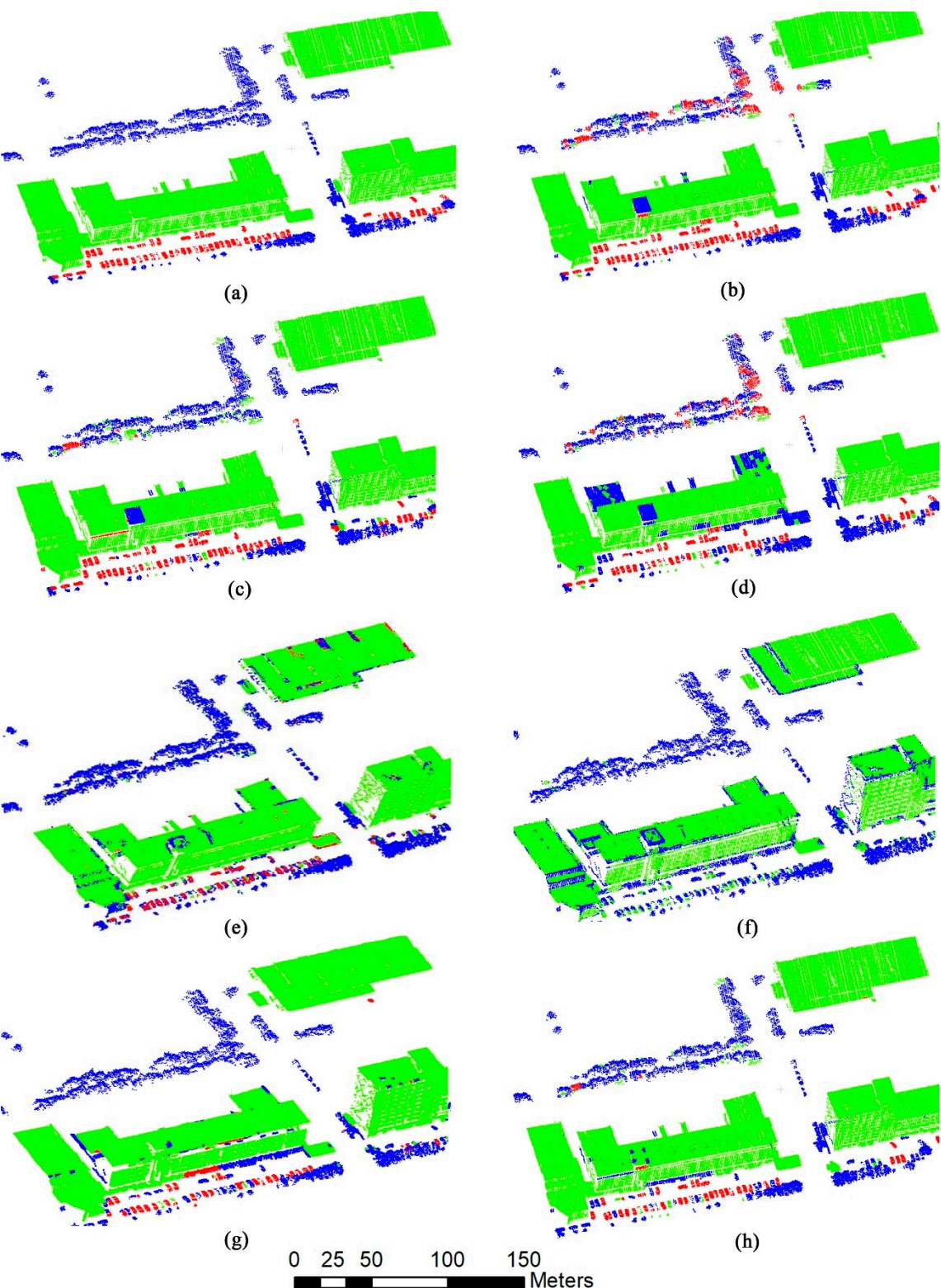

**Figure 8.** The classification results of Scene2. (**a**) Ground truth; (**b**) our method; (**c**) LLC-LDA-SVM; (**d**) LLC-MP-SVM; (**e**) JointBoost; (**f**) AdaBoost; (**g**) BOW-LDA; (**h**) SC-LDA-MP. Note that red, green, and blue points represent cars, buildings, and trees, respectively.

### 5.2.2. MLS and TLS Point Clouds

To verify the applicability of the proposed algorithm to different types of point clouds, Scene3 and Scene4 are tested in this section. The training set and the testing set of experimental data are shown in Table 2. Table 5 shows the classification results of our method and Methods 1–6. From Table 5, our method obtains the highest values on Scene3 and Scene4, reaching 87.5%/77.5%, 60.8%/45.8%, 77.2%/39.6%, and 72.3%/55.9 with regard to OA, mIoU, Kappa, and $mF_1$. It also demonstrates the advantage of the proposed method. For Scene3, our method can achieve the best overall classification performance for the objects with fewer training samples such as poles. For Scene4, the classification performances of all methods are not ideal. Although the LLC-MP features can get the highest $F_1$-score values of pedestrians and cars, our method gets the highest values for other classification metrics. This proves that LLC-MP features have more effective features representation for objects with fewer samples, such as smaller objects of pedestrians and cars. Compared with the classification results of Scene3 and Scene4, the best results of each classification evaluation metrics belong to point set-based classification methods, from which our method achieves the best performance.

**Table 5.** Classification results of Precision/Recall, OA, mIoU, Kappa, and $F_1$-score (%) on Scene3 and Scene4. The best results are highlighted in bold.

| Scene3 | Pole | Building | Car | Tree | OA | mIoU | Kappa | $F_1$-score | $mF_1$ |
|---|---|---|---|---|---|---|---|---|---|
| Our method | 33.1/**36.4** | 90.6/**94.6** | 77.0/**87.4** | 96.4/71.8 | **87.5** | **60.8** | **77.2** | **34.7**/**92.6**/**81.9**/82.3 | **72.3** |
| LLC-LDA-SVM | **47.7**/17.5 | **92.8**/55.9 | 34.1/85.0 | 90.5/**86.0** | 66.5 | 44.8 | 49.3 | 25.6/69.8/48.8/**88.2** | 58.1 |
| LLC-MP-SVM | 24.2/**36.4** | 87.1/93.4 | **77.7**/81.2 | **96.8**/68.9 | 85.6 | 58.1 | 73.3 | 29.1/90.1/79.4/80.5 | 69.8 |
| DKSVD | 1.4/0.8 | 70.3/86.7 | 31.0/4.5 | 58.7/62.7 | 66.4 | 27.9 | 31.8 | 1.0/77.6/7.9/60.6 | 36.8 |
| LCKSVD1 | 3.0/6.0 | 74.6/65.3 | 24.6/13.5 | 42.0/71.4 | 56.7 | 25.3 | 26.3 | 4.0/69.6/17.4/5.9 | 36.0 |
| LCKSVD2 | 5.0/9.6 | 71.2/82.1 | 29.3/4.0 | 50.6/62.0 | 63.5 | 26.8 | 29.2 | 6.6/76.3/7.0/55.7 | 36.4 |
| MSF-SVM | 0.0/0.0 | 72.9/95.7 | 0.0/0.0 | 78.7/77.5 | 74.1 | 33.7 | 44.8 | 0.0/82.8/0.0/78.1 | 42.7 |
| **Scene4** | **Pedestrian** | **Building** | **Car** | **Tree** | **OA** | **mIoU** | **Kappa** | **$F_1$-score** | **$mF_1$** |
| Our method | 72.6/23.7 | **76.4**/**100.0** | **100.0**/7.7 | 77.3/**99.7** | **77.5** | **45.8** | **39.6** | 35.7/**87.1**/14.3/**86.6** | **55.9** |
| LLC-LDA-SVM | **90.5**/14.7 | 23.9/99.8 | 71.3/5.7 | 75.2/81.3 | 63.7 | 27.0 | 22.0 | 25.3/78.1/10.6/38.6 | 38.1 |
| LLC-MP-SVM | 86.0/**28.9** | 58.9/40.4 | 98.2/**13.1** | 75.2/99.4 | 75.4 | 36.8 | 31.1 | **43.3**/85.6/**23.1**/47.9 | 50.0 |
| DKSVD | 8.7/1.3 | 23.9/77.4 | 24.3/1.2 | 74.9/87.2 | 64.9 | 23.0 | 18.3 | 2.3/80.6/2.3/36.5 | 30.4 |
| LCKSVD1 | 10.5/3.8 | 18.2/28.0 | 35.0/6.6 | 73.5/91.0 | 66.1 | 22.4 | 13.6 | 5.6/81.3/11.1/22.1 | 30.0 |
| LCKSVD2 | 12.0/1.7 | 23.6/51.4 | 33.5/0.9 | 74.5/93.4 | 67.8 | 23.1 | 17.5 | 3.0/82.9/1.8/32.3 | 30.0 |
| MSF-SVM | 0.0/0.0 | 35.1/89.4 | 0.0/0.0 | 75.7/94.2 | 70.1 | 26.5 | 24.2 | 0.0/83.9/0.0/50.4 | 33.6 |

In order to more intuitively show the point cloud classification performances of different methods, Figures 7 and 8 show the partial results of different classification methods on Scene1 and Scene2. As shown in Figures 7 and 8, we know that the building classifications of LLC-MP-SVM, JointBoost, and AdaBoost are relatively poor, and there are many misclassifications for building and tree. This constrains the applications of these methods. For LLC-LDA-SVM, BOW-LDA, and SC-LDA-MP, there are still many noise points of different categories. We find that our results are approaching the ground truth. SC-LDA-MP and ours have better classification performance on the building class. They can obtain more complete building contour information. According to the comparisons of Figure 7c,d and Figure 8c,d, the classification method based on LLC-LDA features and the classification method based on LLC-MP features can produce a certain complementarity.

### 5.3. Parameters Sensitivity Analysis

Our method consists of five key parameters, i.e., number of maximum point thresholds $T$ in the point set, number of dictionary words $M$, dictionary learning regular term parameter $\lambda$. local region parameter $K_n$, and number of latent topics $\ell$. In this paper, $T$ is selected from the set: {100, 200, 300, 400}. $M$ is selected from the set {64, 128, 256, 512}. $\lambda$ is selected from the set: {0.0001, 0.0005, 0.001, 0.005, 0.01}. $K_n$ is selected from the set {5, 10, 15, 20}. $\ell$ is selected from the set: {8, 10, 12, 14, 16}.

### 5.3.1. Sensitivity Analysis of Parameter T

To discuss the influence of the point sets generation threshold of the finest layer on the point cloud classification, different number of maximum points in point set $T$ are selected to generate point sets with different sizes. Other parameters are fixed. Point clouds classification experiments are carried out on Scene1 (ALS) and Scene3 (MLS) with $M = 128$, $\lambda = 0.0001$, $K_n = 5$, and $\ell = 16$. The classification results of different thresholds $T$ are shown in Figure 9. For Scene1, the $F_1$-score values of trees and buildings are more than 90%, and the $F_1$-score values of cars are more than 50%, as shown in Figure 9a. According to Figure 9a, we can see that $T$ has a certain influence on the classification effects. However, when the $T$ is selected at the appropriate range, there is relatively little influence on the point cloud classification. As shown in Figure 9b, for Scene3, when $T = 100$, 200, and 400, the gaps of OA, mIoU, and Kappa are less than 5%. For the $F_1$-score values of each category, the larger maximum point number of the point set in the finest layer is set, which obtains worse classification performance of the poles. Besides, the other categories have relatively little difference. According to Figure 9a,b, except for the case of $T = 300$, the gaps of overall evaluation metrics in other cases are less than 5%. Therefore, the number of maximum points in the point set of the finest layer has a relatively small impact on the point cloud classification. In addition, shape, density, and size of the single objects, which belong to the same kind of category in Scene1, have slight differences, while those in Scene3 have great differences (e.g., poles and trees). As shown in Figure 9, $T$ has a relatively large influence on cars and poles. This is because there are few cars/poles samples for training, and the number of points in the point sets of cars/poles is relatively small. For relatively large objects such as buildings and trees, the classification performance is improved when the size of point set in the finest layer increases within a certain range. For point clouds in different scenes, the threshold $T$ can be adjusted according to the density and the shape of the objects in the point cloud. Thereby, the local information of the objects can be adequately expressed by the finest layer point sets, and enough object points can be ensured in the point sets.

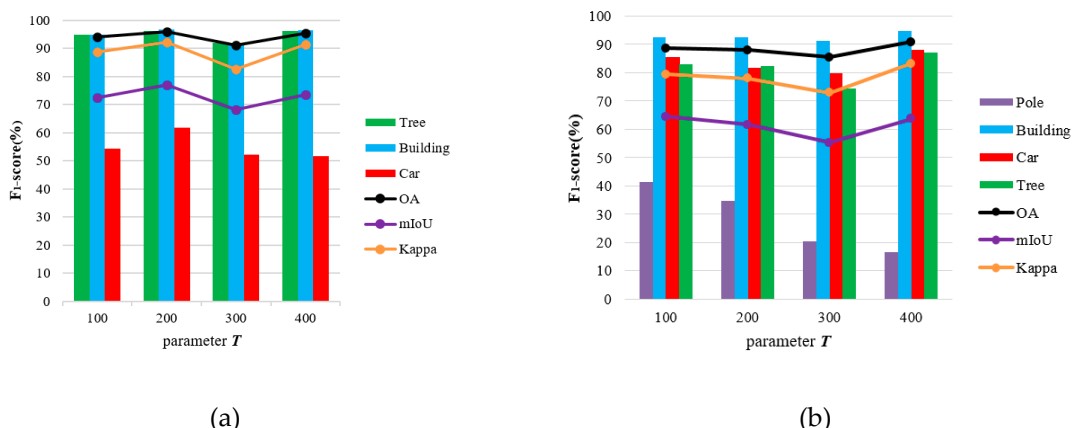

(a)                                                                 (b)

**Figure 9.** Point cloud classification performance with different numbers of maximum points in the point sets. (**a**) Scene1; (**b**) Scene3. Among them, the line charts are the curves of the overall evaluation metrics, i.e., OA, mIoU and Kappa (%), and the histograms are the $F_1$-score values (%) of each category classification.

### 5.3.2. Sensitivity Analysis of Parameters on Dictionary Learning and Sparse Representation

The number of dictionary words, the regular term, and the local region range are important parameters for dictionary learning and sparse representation. To evaluate the effects of these parameters in dictionary learning and sparse representation for point cloud classification, we implement an experiment using Scene1 and Scene3 datasets. Firstly, we fix $T = 200$, $\lambda = 0.0001$, $K_n = 5$, and $\ell = 16$. We use the different values of dictionary words $M$ (e.g., 64, 128, 256, and 512) to test the accuracy of classification results, as shown Figure 10. As shown in Figure 10a, the $F_1$-score values of the building

show an upward trend with the number of dictionary words increasing. For trees and cars, the classification performance can be improved when the number of dictionary words is within a certain range. When it exceeds the appropriate range, the classification performance may be poor. For the overall evaluation metrics, the changes of the number of dictionary words have less influence on OA but have great influence on classification consistency (Kappa) and mIoU. For Scene3, Figure 10b shows that, when the number of dictionary words changes, the values of overall evaluation metrics, i.e., OA, mIoU, and Kappa, are changed slightly (less than 5%). However, the number of dictionary words has a relatively large impact on the classification of poles and trees. The influence trend of the dictionary words number is the same as Scene1. To this end, when the number of dictionary words is at the range of 128~256, our method can achieve relatively good classification results. In addition, for categories with fewer samples, e.g., cars and poles, the performance of point cloud classification is greatly influenced by the number of dictionary words.

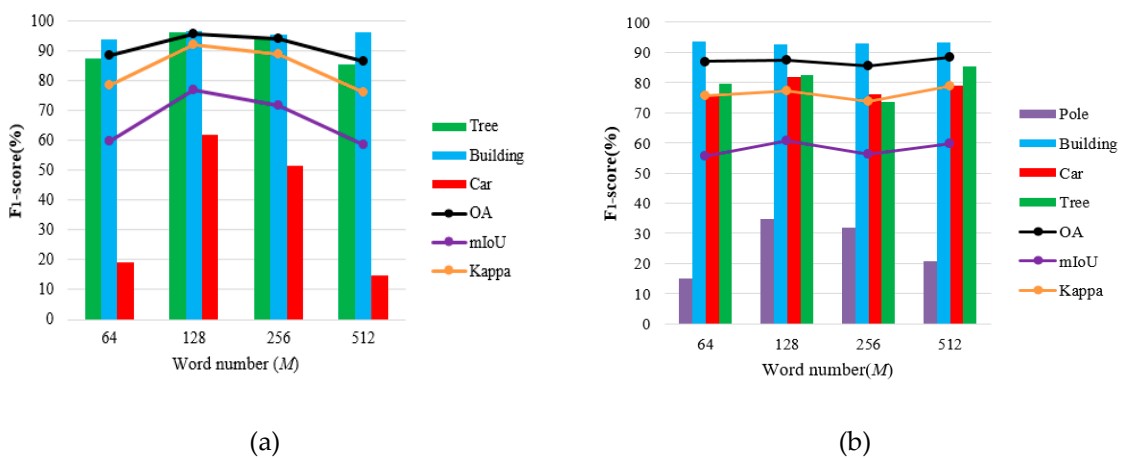

(a)                                                    (b)

**Figure 10.** Point cloud classification performance with different numbers of dictionary words. (**a**) Scene1; (**b**) Scene3. Among them, the line charts are the curves of the overall evaluation metrics, i.e., OA, mIoU, and Kappa (%), and the histograms are the F1-score values (%) of each category classification.

The regular term parameter $\lambda$ and the local region parameter $K_n$ are often coupled with each other and have an impact on dictionary learning and sparse representation. To test their impact on classification accuracy, we set $T = 200$, $M = 128$, $\ell = 14$ and 16. The result is shown in Figure 11 when parameter $\lambda$ is set to 0.0001, 0.0005, 0.001, 0.005, and 0.01 and $K_n$ is set to 5, 10, 15, and 20, accordingly.

For Scene1, as shown in Figure 11a,b, the number of latent topics is 14 and 16, and when $K_n$ is chosen in the range [5,10], the value of metric OA can achieve a relatively good performance. When $K_n$ is greater than 10, OA tends to decline. When $\lambda$ is at the range of 0.0001~0.05 with $K_n$ at the given range, the OA of the point cloud classification shows a downward trend. However, when $\lambda$ is 0.1, the OA of the point cloud classification has a certain increase, while there is still a certain gap compared with the OA with $\lambda = 0.0001$. As shown in Figure 11c,d, it can be seen that the mIoU has similar distribution trends with OA, but the difference of classification performance caused by the change of each parameter for mIoU is more obvious than OA, which is mainly due to the parameter sensitivity for the small samples classification.

For Scene3, as shown in Figure 11e,g, the changes of $\lambda$ and $K_n$ have little influence on the OA, and the general trend is similar to that of Scene1. From Figure 11f,h, we can see that the mIoU is obviously affected by the parameter $K_n$, and the larger $K_n$ is, the lower mIoU value is. However, mIoU is less affected by $\lambda$, and the overall trend is similar to that of Scene1.

The above comparative analysis demonstrates that, when $K_n$ and $\lambda$ are set at the range of 5~10 and 0.0001~0.0005, respectively, promising point cloud classification performance can be obtained.

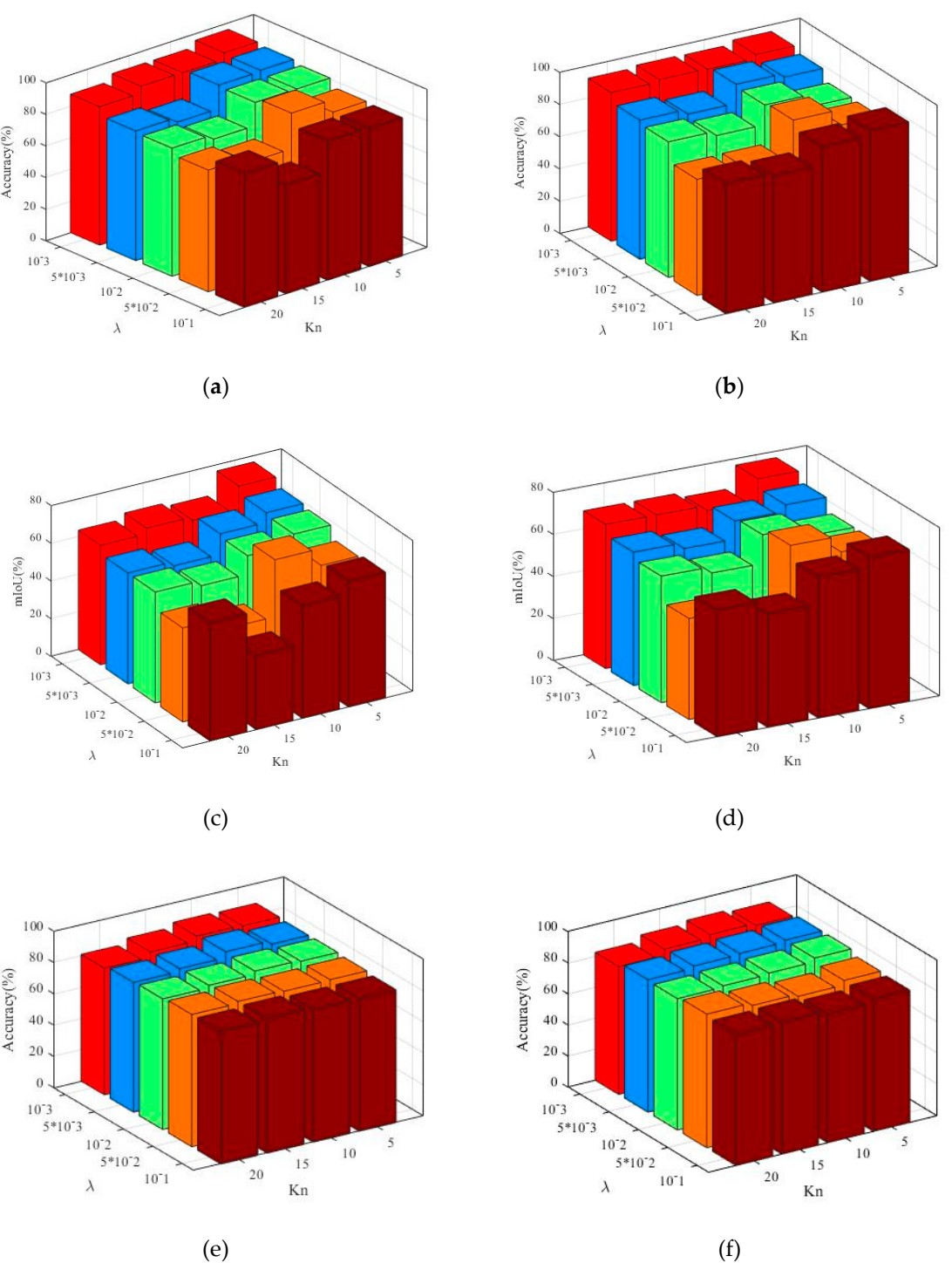

(**a**)             (**b**)

(c)             (d)

(e)             (f)

**Figure 11.** *Cont.*

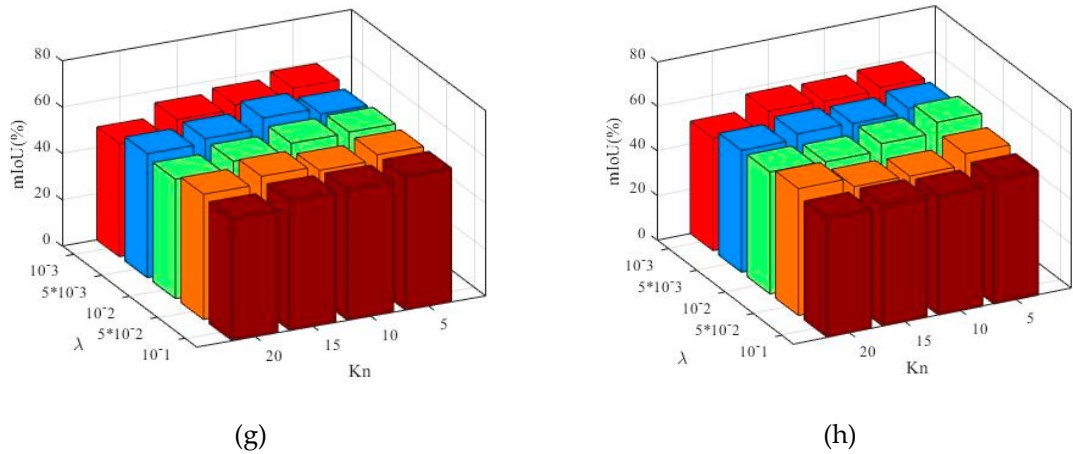

<div align="center">(g)　　　　　　　　　　　　　　　　　　　　(h)</div>

**Figure 11.** Point cloud classification performances with different trade-off parameter values and local region range parameter values in dictionary learning and sparse representation. (**a**) and (**b**) are different OAs obtained with different $\lambda$ and $K_n$ on Scene1 when the number of latent topics are 14 and 16. (**c**) and (**d**) are different mIoUs obtained with different $\lambda$ and $K_n$ on Scene1 when the number of latent topics are 14 and 16. (**e**) and (**f**) are different OAs obtained with different $\lambda$ and $K_n$ on Scene3 when the number of latent topics are 14 and 16. (**g**) and (**h**) are different mIoU obtained with different $\lambda$ and $K_n$ on Scene3 when the number of latent topics are 14 and 16.

### 5.3.3. Sensitivity Analysis of Latent Topics Number $\ell$

The number of latent topics determines the dimensions of the point set LLC-LDA features. In order to discuss the influence of the latent topics number on the point cloud classification, we set $T = 200$, $M = 128$, $\lambda = 0.0001$, and $K_n = 5$, and select 8, 10, 12, 14, and 16 latent topics to conduct experiments, respectively. Point cloud classification experiments are performed on Scene1 (ALS) and Scene3 (MLS). The influence of different latent topic numbers on point cloud classification is shown in Figure 12. As shown in Figure 12, when the different latent topics number $m$ is selected, the classification accuracy of some categories has a certain degree of difference. It is to note that the proportion of these objects in training samples is very small, i.e., taking up to 4.8% and 2.8% of all training points, respectively. Therefore, the latent topics number affects the classification performance of categories with fewer samples. However, as shown in Figure 12, the overall classification metrics, i.e., OA, mIoU, and Kappa, have little difference (less than 5%) with various latent topics numbers. We conclude that the latent topics number at the range of 8~16 has a relatively small impact on the point cloud classification.

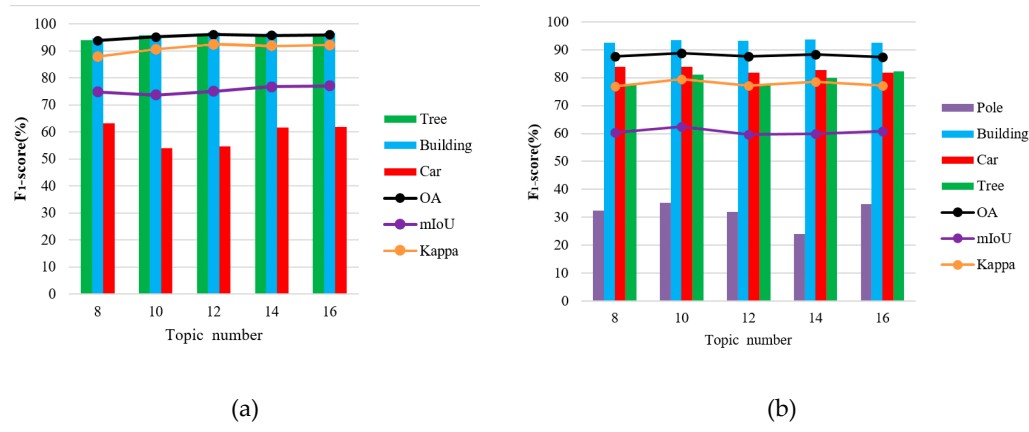

<div align="center">(a)　　　　　　　　　　　　　　　　　　　　(b)</div>

**Figure 12.** Point cloud classification performances with different latent topics numbers. (**a**) Scene1; (**b**) Scene3. Among them, the line charts are the curves of the overall evaluation metrics, i.e., OA, mIoU, and Kappa (%), and the histograms are the F1-score values (%) of each category classification.

## 6. Conclusions

This paper presents a novel point set features extraction method via multi-level global and local features aggregation for point cloud classification. The proposed method firstly generates different levels of point sets by means of multi-level clustering. The point sets of each level have different sizes, which can express the different parts and structures of objects. In this step, we provide robust and significant point set features. Afterwards, the LLC-LDA and the LLC-MP multi-level aggregation features of the point set are extracted based on the covariance eigenvalues features and the spin image features. In the point set features extraction, LLC-based dictionary learning and sparse representation are used to make full use of the locality between each neighboring point, which makes the sparse representation more significant. Finally, point clouds can be classified based on multi-level aggregation features of point sets followed by fusing the global and the local information representations of different hierarchical point sets, i.e., LLC-LDA features and LLC-MP features. The experimental results show that the multi-level point set features extracted by our method are significantly discriminative, and the extracted features can effectively express different types of complex objects. Moreover, the point cloud classification of our method outperforms other comparison algorithms in most evaluation metrics.

Although our method achieves more accurate classification results in the data set shown in Table 2, our method still has certain drawbacks. (1) As the more robust and significant point set multi-level aggregation features and classification model need more training samples to generate, larger datasets with labels are needed, which requires more time in labeling the data and training the model. (2) Integration of local and global features by simply concatenating them together does not achieve optimal results for the expression of these two types of features. How to effectively integrate features from different perspectives is the focus of future works. In addition, based on the proposed framework, combining deep learning methods for features extraction and fusion is also an improvement direction for future research.

**Author Contributions:** Conceptualization, Y.L.; methodology, Y.L., and Z.Z.; software, Y.L., J.Z. and W.Z.; validation, G.T., Y.L., J.Z. and D.C.; data curation, D.C., W.Z. and J.Y.; writing—original draft preparation, Y.L., J.Z. and W.Z.; writing—review and editing, Z.Z., D.C. and W.Z.; supervision, G.T.; project administration, G.T.; G.T., Y.L. and D.C. contribute equally to the manuscript.

**Funding:** This work was jointly supported by the National Natural Science Foundation of China under Grant 41971415 and Grant 61175031.

**Acknowledgments:** The authors would like to thank Qi Sun at Northeastern University for manuscript revision.

**Conflicts of Interest:** The authors declare no conflict of interest.

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
