# Peer review of "Point Set Multi-Level Aggregation Feature Extraction Based on Multi-Scale Max Pooling and LDA for Point Cloud Classification"

_remotesensing, doi:10.3390/rs11232846_

Round 1

Reviewer 1 Report

The paper analyzes the possibility of introducing an algorithm for the classification of point clouds, with the help of a comparative study. The basic constructive characteristics of the used equipment are missing. This is extremely useful to be able to observe the standard position error in the initial take-over of points in the field. I also propose a clearer restoration and presentation of figures 2a-d, 5a-d, 8a-d, because, from the authors' presentation, the data cannot be correctly interpreted. Overall, the paper presents an interesting case study, with scientific research potential, which must be restored in the proposed form in order to be accepted for publication.

Author Response

Kindly see the attached file.

Reviewer 2 Report

This research subject fits within the scope of Remote Sensing Journal and is an interesting topic and the presented point cloud classification probably will have real application. The paper is almost suitable for publication, requires a minor revision.

The paper is well structured and prepared according to instructions for authors.

There are some points that might be better:

Figure 1, the flow chart is a bit too complex, it has many elements. My suggestion (not necessary) is to make one simple basic flow chart and describe it by parts Table 2 – it is missing the points (Scene1 68,802 – it is not written what it is all about) Figures 4-8 – it is missing scale, please add. Also I would like to know some basic facts about tested point cloud (area, point density….) Table 3, please add references to compared methods, it would be good if they were seen in the table and not just in the previous text

Is it possible to discuss, how the described method would perform larger (real) point clouds? This makes me a question about this paper a bit, the algorithm is developed and tested on the relatively small point clouds.

Author Response

Kindly see the attached file.

Reviewer 3 Report

The article presents a point cloud classification method.
First, the point cloud is segmented using standard methods, i.e., DBSCAN and recursive K-means.
Then, both global and local features are extracted from each segment, using proposed methods LLC-LDA and LLC-MP.
The features are used to classify the segment using a SVM.
The proposed method was compared against other methods in literature, and it out-performed them in many cases.

The method proposed in the article is derivative, as the proposed pipeline is composed of well-known algorithms.
However, the pipeline itself seems novel, and the experimental evaluation involves comparison with many competing algorithms.

The authors should also report computation times for all compared methods, not only for the proposed method (line 410), to highlight the tradeoff.

It seems that the dataset does not include roads and sidewalks. Please add a short explanation about how they were removed from the point cloud.

The experimental evaluation only considers three or four classes. The scenarios did not contain any other type of object?

The writing is acceptable. Some unclear parts and/or typos:
Line 88: "After being implemented DBSCAN"
Line 195: This algorithm is unclear. Please use a simpler and more concise pseudocode.
Line 195: Why is a table caption (Table 1) above the algorithm?
Line 197: What is displayed in Fig. 2d? Is it the ground truth?
Line 205: Please provide a zoomed-in detail of some part of this figure. Segments in Fig. 2c are too small.
Line 233: What is the relation between f_i and F_{cov}?
Line 262: "normalize all single point features F_{m-point} over each column" Please clarify the meaning of "column" here.
Line 293: "needs to be set zero" missing "to"
Line 394: The numbers in Table 2 are points, as specified in the column headers, or point clouds, as stated at line 392?
Line 407: The phrase "software package of" is redundant.
Line 480: "Compared with" It is unclear what is being compared with what.
Line 481: "performance of cars"
Line 584: "Sence" (twice)
Line 597: "have greatly"

Author Response

Kindly see the attached file.

Reviewer 4 Report

Page 3, Line 125, Figure 1:

The flowchart on Figure 1 shows LLS, LDA, but MP is not marked clearly (I assume it means “multi-scale maxpooling”).

Page 4, Line 155, Line 165:

“...position relationships and point number.”

“...number of point set.”

“point number” – is this some special number or do you mean a number of points/ the size the point set? Actually, you also use “number of points” in other parts of the paper. Thus, it is a little confusing. Please check and possibly correct this issue on other parts of the paper.

Page 5, Line, 188, Line 195, Algorithm 1:

You set K=2. What is exactly is this parameter? In Algorithm K is the number of clusters centers. Thus, you always have only two cluster centers? You also select K points (“is randomly selected, and K points are selected as the initial centroid”). This is a little confusing. Please check, if a particular parameter needs to be named differently.

Page 10, Line 391:

“...at the website: http://geogother.bnu.edu.cn/teacherweb/zhangliqiang/.

You should put the website url in the footnote or as the reference.

Page 12, Lines 427-430, Table 3:

You compare your method with Method 1 (LLC-LDA-SVM) and Method 2 (LLC-MP_SVM). You did not provide any reference for these two methods. Moreover, these two methods use your point sets construction method and your proposed LLC-LDA (LLC-MP) aggregation features for point classification method.

Thus, this is a little confusing. Are Method 1 and Method 2 your previous work and also applied the approaches described in this paper. Please explain.

Page 18, Line 563:

It also demonstrates the superiority of the proposed method.“

Superiority” is a strong word. Of course, some results of your method are better. Nevertheless, how do you justify the use of this strong word?

Page 18, Line 577:

...the number of latent topics m. “

You marked the number of latent topics as “m”, but later in the paper, you marked this parameter as small letter “l” (Page 18, Line 579 Page 22, Line 670, Section 5.3.3).

Author Response

Kindly see the attached file.
